# Potential airborne transmission of SARS-COV-2 through bathroom ventilation ducts associated with an outbreak in a residential building in Santander, Spain, 2020

Shelly L. Miller[ID][1]*, Shujie Yan[2], Alberto García[1], Liangzhu Leon Wang[ID][2], Zhiqiang Zhai[3], Jose Ramon Aranda[4], Fernando González-Candelas[ID][5], Ignacio Lombillo[ID][4], Javier Balbás[ID][4], Ernesto Cabrillo[6], Delfín Silió[7], L. David Higuera[6]

**1** Mechanical Engineering, University of Colorado Boulder, Boulder, Colorado, United States of America, **2** Centre for Zero Energy Building Studies, Civil, and Environmental Engineering, Concordia University, Montreal, Canada, **3** Civil, Environmental, and Architectural Engineering, University of Colorado Boulder, Boulder, Colorado, United States of America, **4** Building Technology R&D Group (GTED-UC) Construction Technology Group, Universidad de Cantabria, Cantabria, Spain, **5** Foundation for the Promotion of Health and Biomedical Research, Valencian Community – Fisabio and Institute for Integrative Systems Biology (I2SysBio), University of Valencia - CSIC (Valencia, Spain), Valencia, Spain, **6** Higuera Engineering Technical Office (Santander, Spain), Cantabria, Spain, **7** Electrical and Energy Engineering, Universidad de Cantabria, Cantabria, Spain

* Shelly.Miller@Colorado.EDU

## Abstract

During the COVID-19 pandemic, airborne transmission of SARS-CoV-2 via respiratory aerosols was a critical concern in indoor environments. In the city of Santander, Spain, an outbreak in a multi-family residential building during a period of low community transmission revealed vertical clustering of 15 cases in four homes. The building's design included single interior bathrooms without windows in each home, ventilated by a shared vertical bathroom duct system. Field measurements, computational fluid dynamics (CFD) simulations, and multi-zone airflow modeling were performed to evaluate vertical disease transmission potential in the Santander building. Epidemiological and genetic data combined with the field-collected data and modeling indicated that the most plausible transmission route was the bathroom vertical ventilation duct system, which facilitated movement of infectious aerosol between vertically connected homes. Additionally, operating the kitchen exhaust fan can augment the movement of aerosols between occupied spaces increasing the potential for infection. Recommendations for mitigating future risks include the installation of forced air exhaust fans with non-return flaps in bathroom ducts.

## Introduction

SARS-CoV-2, the causative agent of COVID-19, is primarily transmitted by individuals infected with the virus exhaling viable aerosols containing the virus [1]. These

**Data availability statement:** Data and supporting documents are held in the following repository: https://scholar.colorado.edu/concern/articles/br86b5432.

**Funding:** The author(s) received no specific funding for this work.

**Competing interests:** The authors have declared that no competing interests exist.

respiratory aerosols can remain suspended in the air and travel with the air currents, potentially infecting individuals who inhale them at short and long distances from the infected source, especially in insufficiently ventilated indoor spaces [2].

In addition to transmission in shared indoor spaces, virus spread without close contact has been documented. In 2003, a SARS-CoV-1 outbreak in Hong Kong's Amoy Gardens infected around 320 people in separate homes [3]. In Seoul (2020), COVID-19 cases clustered along two vertical shafts of a multi-family building, linked by a single bathroom ventilation duct lacking individual extractor fans [4]. Similarly, a 2022 Hong Kong study found that 8.7% of residents in high-rise buildings were affected by Omicron due to vertical transmission [5].

In the city of Santander, Spain, a SARS-CoV-2 outbreak occurred in the early summer of 2020 in a seven-story residential building with 56 homes. Following Spain's partial lockdown and subsequent easing of restrictions, a sudden cluster of 15 cases emerged in four vertically stacked homes connected by a shared bathroom ventilation duct. This clustering happened during a period of almost zero community transmission, suggesting an internal transmission pathway.

This study investigates how air moves through shared vertical bathroom ventilation ducts in bathrooms and windows of the Santander building, evaluating its role in the transmission of SARS-CoV-2 aerosols between homes. Prompted by a building resident – an engineer – who identified the issue, the study was led by a multidisciplinary international team of professors and students from the University of Valencia, the University of Cantabria, the University of Colorado Boulder, and Concordia University.

## The study

The investigation into the COVID-19 outbreak linked to bathroom ventilation ducts in the Santander residential building began with formal approval from the building's homeowners' association. Residents supported the study (see Supplemental Information, S1-S4 File). The University of Valencia research team collaborated with the Regional Ministry of Health to obtain anonymized outbreak data, while additional epidemiological details were sourced from official statements and media reports due to limited online access. The outbreak was first identified by a resident and co-author of this paper, David Higuera, who noted a cluster of cases on his floor; subsequent reports revealed that the second affected apartment was directly above the first, aiding in tracing the spread.

To understand how the outbreak could have happened, the research team then conducted a comprehensive study that included indoor airflow measurements in a typical home, a simplified CFD model, and 3D multi-zone airflow simulations to evaluate aerosol movement between homes. The study also presents details about the outbreak, the building characteristics, and testing for COVID-19. Based on these findings, technical solutions are proposed to mitigate vertical disease transmission, along with recommendations for updates to Spain's building inspection standards.

### Outbreak in the Santander building

A COVID-19 outbreak occurred in a building in Santander after Spain imposed a partial lockdown starting March 14, 2020. On March 30, a country-wide two-week

mandatory confinement period for all non-essential workers was implemented. Lockdowns began easing on April 28, and the state of alarm ended on June 21. Fig 1 illustrates the timeline for the number of confirmed COVID-19 cases in Santander starting on March 20, and by June, cases had dropped to zero. [6]. On June 21, the regional health authority detected an outbreak of COVID-19 in the Santander building. Fifteen (15) people became infected in four different homes vertically stacked [7]. The outbreak began suddenly and rapidly after a period of virtually no community transmission of the virus in the city (population 172,000).

## Characteristics of the building

Constructed in 1969, the building comprises seven floors with eight homes per level, grouped around four patios (Fig 2a). The building was constructed in 1969; it predates the 1970s Technological Standards and Basic Building Standards in Spain [8]. All common areas, including the elevators, are used by all building residents. Each home has a single interior bathroom (without a window). The bathrooms were originally built with natural convection exhaust ventilation via a construction opening on the vertical wall behind the toilet near the ceiling. This opening, with dimensions of 12 x 12 centimeters (cm), is connected to a vertical bathroom duct measuring 70 x 20 cm that serves as a conduit for natural convection airflow to the roof of the building. This bathroom ventilation duct is shared by pairs of homes with the same patio (Fig 2b), and it also serves as a conduit for the wastewater and sewage pipes. Some homes had modified their ventilation by installing extraction fans or blocking the exhaust due to odor concerns before the outbreak.

## Testing

On June 21, 2020, the regional health authority detected the outbreak through polymerase chain reaction (PCR) testing, initially in a third-floor home where all six inhabitants tested positive (Fig 3). On June 25, a positive PCR test confirmed that COVID-19 was detected in the home immediately above, on the fourth floor, and all three inhabitants of this home were infected [9]. On June 26, the health authorities carried out a general PCR screening of the entire building's occupants (some homeowners were away on travel) that yielded four new positive cases. The next day, general confinement of the building for 97 residents was mandated for ten days. Unaffected homes may have had prior immunity, but this remains unconfirmed. No public report fully explained the vertical clustering; possible contributing factors such as lapses in hygiene or mask use were noted, but no direct contact between non-cohabitants was identified.

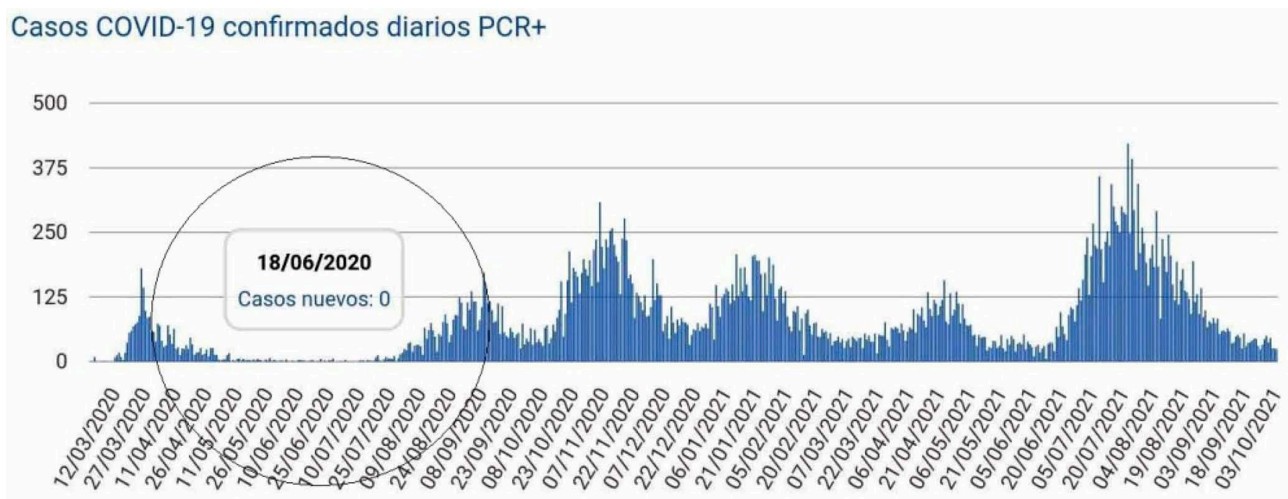

**Fig 1. Timeline of confirmed COVID-19 cases in Santander from March 2020 through October 2021.**

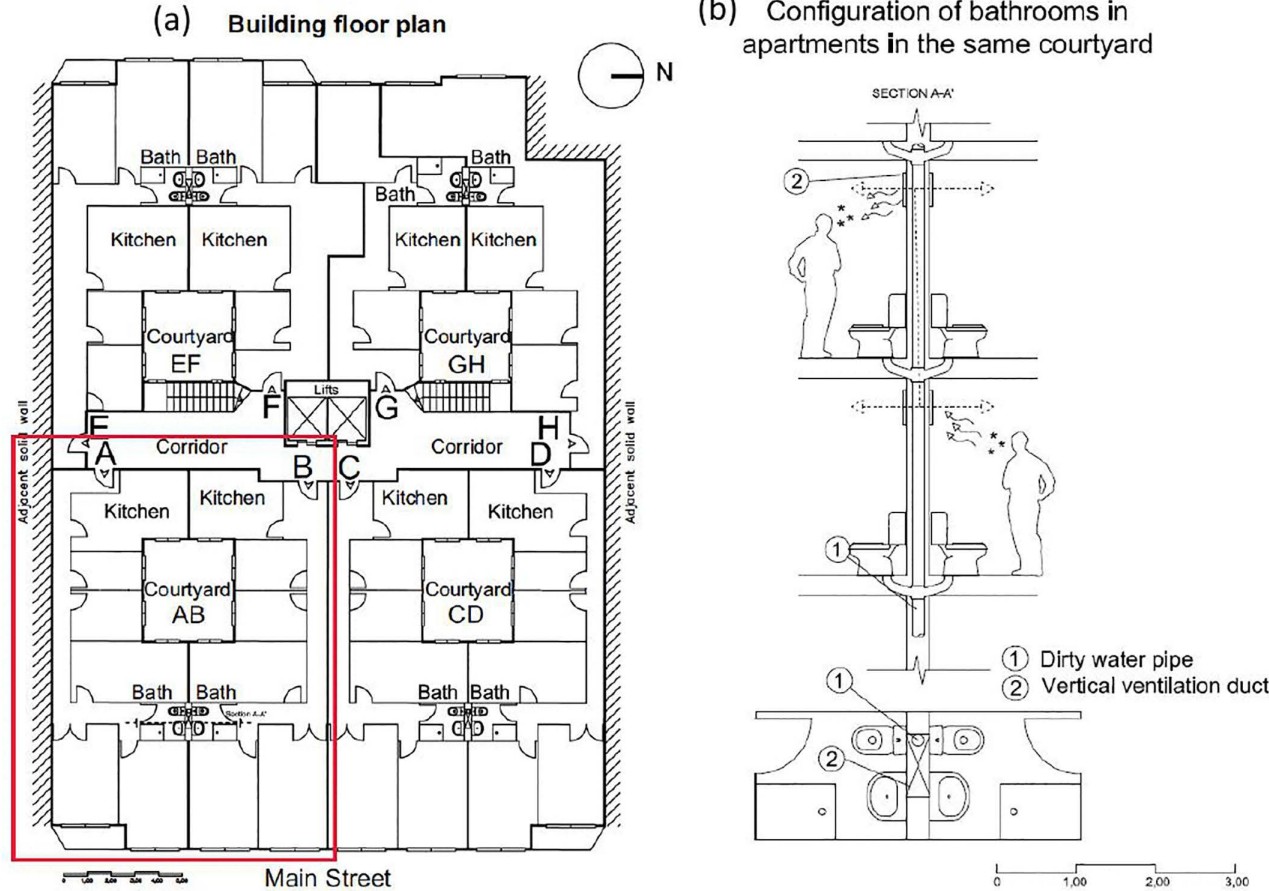

**Fig 2. Plan view of one of the floors of the multi-family housing building in Santander (a), and vertical diagram of the natural ventilation opening of the bathroom and the duct that connects to the roof (b).** There are four homes on each floor, grouped two by two around four patios. The red box represents the affected homes and patio that share the same vertical bathroom ventilation duct.

The health officials collected samples of the surfaces in the common areas and tested them for SARS-CoV-2. However, all these samples were negative. Authorities also ordered the surfaces of the common areas of the building to be disinfected. Additionally, approximately 500 PCR tests were performed on three consecutive dates on close contacts visiting the building (grandparents, cousins, friends, etc.), occasional visitors (postman, cleaner, maintenance staff), parish priests, shopkeepers, and hoteliers, all with negative results. No positive PCR tests were detected among the residents of the homes surrounding the three other patios of the building. The lack of infection around the other courtyards is likely due to the absence of an Index Case in those homes. PCR tests were conducted in the whole building (not only to the A and B sections of the building; Fig 2).

On July 1, a second round of PCR testing was conducted among building occupants, detecting two new positive cases. Most individuals experienced only mild symptoms; however, one elderly woman required hospitalization. On July 8, during the third and final round of PCR testing, residents who tested negative were released from confinement. Ultimately, 15 cases were identified in four homes along the same bathroom duct (Fig 3). No residents from other parts of the building were affected [10], although it remains unknown whether the unaffected homes (connected to the same bathroom duct) had developed immunity from prior exposure. Interestingly, the occupants of three homes (floors 1, 3, 4) in which the bathroom ventilation had been modified (by installing an exhaust fan with a no-return flap) did not get infected (Fig 3).

**Fig 3. Side and three-dimensional view of the Santander building.** The index cases were on the 3rd floor-Home A (dark red), testing positive on Jun 21, 2020. Subsequent infections (light red) occurred on floor 1-Home B, floor 4-Home A, and floor 6-Home A. Letters A and B indicate that the infection occurred in Home A or Home B on that floor. Homes A and B shared a vertical bathroom duct and patio area. The other 10 homes sharing the vertical bathroom duct were either inhabited or not occupied, or their bathroom duct was modified, and no infection was documented.

Occupants in a fourth home (5th floor) that had the bathroom exhaust vent covered did not get infected either. The contagious potential is defined as C/I, where C is the number of new cases, excluding the index cases, and I is the number of infectors. An outbreak is considered possible when C/I exceeds one [11]. In this case, there were 15 total cases, minus the 6 infectors, resulting in nine new cases. The contagious potential is 9/6 = 1.5, indicating an outbreak.

## Genetic sequencing of the virus

Complete genome sequencing was performed on three samples from building residents. These were compared with 16 other samples collected from the same locality around the same time, as well as with the original SARS-CoV-2 Wuhan-1 reference sequence. The protocols of the Spanish Seq COVID-19 consortium [12] were used to obtain a maximum likelihood phylogenetic tree with sequences in Fig 4. The sequences derived from the building were similar, with one or two nucleotide differences among them and at least 11 differences from the closest control sequence. This level of genetic similarity along the complete genome of SARS-CoV-2 isolates is indicative of a recent shared ancestry, either from a common source or through direct transmission.

## Potential transmission routes

Potential transmission routes included shared spaces (elevators, lobbies, or corridors) and surfaces (elevator keypads, handrails, or portal door handles), although this form of transmission of COVID-19 is minimal, with few cases attributed to contact with contaminated surfaces [13]. Fifty-six homes share two elevators; if transmission had occurred via these common areas, infections would likely have appeared in multiple parts of the building. However, all cases were clustered

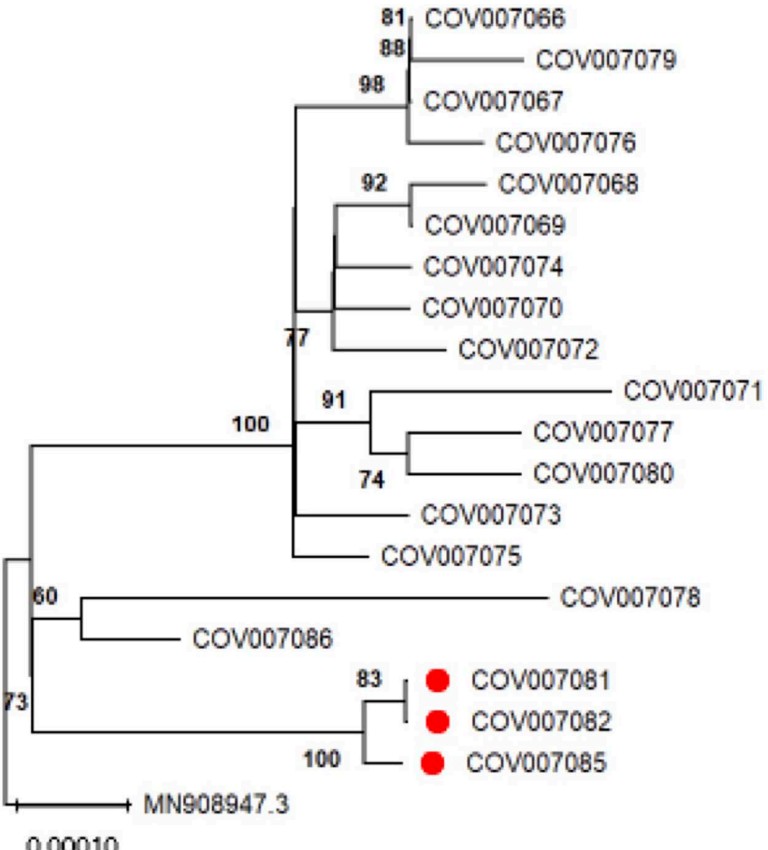

**Fig 4. Maximum likelihood phylogenetic tree obtained from complete genome sequences of SARS-CoV-2.** Sequences derived from samples of inhabitants with COVID-19 in the Santander building outbreak are labeled COV007081, COV007082, and COV007085 and marked with a red dot. The Wuhan-1 sequence is labeled as MN908947.3. The remaining sequences correspond to local controls from Santander taken in June and July 2020. Bootstrap support values higher than 70% are shown in the corresponding nodes. The scale bar represents inferred substitutions/site.

in the same vertical section around a single patio. At the time of the outbreak, mask use was mandatory in the presence or proximity of non-cohabitants [14].

The building's bathrooms were originally designed to exhaust air by natural convection through a vertical ventilation duct leading to the roof – a flow phenomenon known as the *chimney* or *stack effect,* driven by temperature-induced differences in air density. However, several factors can disrupt this intended exhaust airflow, including exhaust hood operation, window openings, and pressure changes within the building. These disruptions can cause reverse flow, where air from the bathroom ventilation duct re-enters the bathroom instead of being expelled to the outside. It is hypothesized that infectious aerosols from one home may have traveled through this shared bathroom duct system and entered other homes via reverse flow, potentially serving as a mechanism for cross-home transmission.

## Measurements

Environmental measurements were collected in the 4th-floor bathroom and the bathroom ventilation duct of Home B (Fig 3), between May and December 2022. Differential pressure was measured between the duct and the bathroom interior (DG1000 Minneapolis, MN), and airspeed was measured at the bathroom opening (TSI VelociCalc 9565, Shoreview, MN). Bathroom concentrations of carbon dioxide ($CO_2$), bathroom interior temperature, and relative humidity (Aranet 4, Riga,

Latvia) were also measured. Additional measurements in other homes were not possible due to a lack of access. Five measurement campaigns were completed to document the behavior of the bathroom exhaust duct under different conditions. Table 1 summarizes the data collected during these campaigns.

The following conditions were explored: operating the kitchen exhaust hood, opening the patio windows, and opening the windows that face the exterior of the building and the street. At the bathroom ventilation exhaust opening, the flow direction was measured as either reverse flow (into the bathroom) or exhaust flow (out of the bathroom), depending on whether the differential pressure between the interior of the bathroom and the duct was positive or negative, respectively. This study was conducted with the permission of the building administration, the homeowner's community, and the collaboration of the residents.

Reverse flow into the bathroom was observed when positive pressures were measured, consistently accompanied by elevated $CO_2$ concentrations when the home was unoccupied, suggesting that $CO_2$ was entering from other homes. These reverse flow events also coincided with increases in humidity and temperature. In contrast, negative pressures were associated with lower $CO_2$ levels and reduced humidity in the bathroom.

## CFD model of airflow rates and $CO_2$ between two bathrooms

A CFD simulation was used to study the possibility of aerosol transmission between two homes' bathrooms from reverse flow caused by pressure differences, using the measurements from the 4th-floor home as boundary conditions. Note that because "stack effect" pressure differentials vary by height, the infiltration rates on the 1st or 2nd floor might differ in magnitude from the measured 4th floor, though the *direction* of flow (reverse flow) likely remains consistent under the observed conditions.

Fig 5 illustrates a setup consisting of two vertically aligned, adjoining rectangular bathrooms (1610 × 1900 mm floor area, 2700 mm height) connected by a vertical bathroom ventilation duct measuring 200 × 700 mm. The model focused on airflow and $CO_2$ distribution across bathrooms on different floors, with some non-essential features simplified to reduce

**Table 1. Measurements Conducted in the 4th Floor Apartment of the Santander Building in 2022.**

| Date (2022) | June 10 | May 1415 | | July 7 | | September 24 | | December 6 | |
|---|---|---|---|---|---|---|---|---|---|
| **Case** | 1 | 2 | | 2 | | 2 | | 3 | |
| **Duration** | Spot measurements | 48 h 45 min | | 24 h | | 24 h | | 230 min | |
| **Occupancy** | N/A | Empty | | Empty | | Empty | | Empty | |
| **Condition** | Open window to the main street | Open window to the community patio | | Open window to the community patio | | Open window to the community patio | | Operating the kitchen hood using fan position 3 (high) | |
| **Carbon Dioxide** | | | | | | | | | |
| **CO₂ Minimum (ppm)** | 550 | 582 | | 558 | | 529 | | 640 | |
| **CO₂ Maximum (ppm)** | N/A | 1489 | | 1032 | | 725 | | 1450 | |
| **% Time < 700 ppm CO₂** | 100% | 61% | | 74% | | 83% | | 4.8% | |
| **% Time > 700 ppm CO₂** | 0% | 39% | | 26% | | 17% | | 95% | |
| **Pressure and Air Flow** | Spot measurements | Average | Max | Average | Max | Average | Max | Average | Max |
| **Natural Convection ΔPa < 0 (Pa)** | −0.6 | N/A | N/A | −0.19 | −1.02 | −0.34 | −1.05 | −0.45 | −1.62 |
| **Flow (l/s)** | −8.3 | N/A | N/A | −4.47 | −11.76 | −4.74 | −11.76 | −17.03 | −43.29 |
| **Air Reverse Flow ΔPa > 0 (Pa)** | N/A | N/A | N/A | 0.2 | 1.01 | 0.21 | 0.91 | 7.29 | 8.03 |
| **Flow (l/s)** | N/A | N/A | N/A | 5.09 | 11.68 | 4.23 | 9.87 | 42.04 | 44.46 |

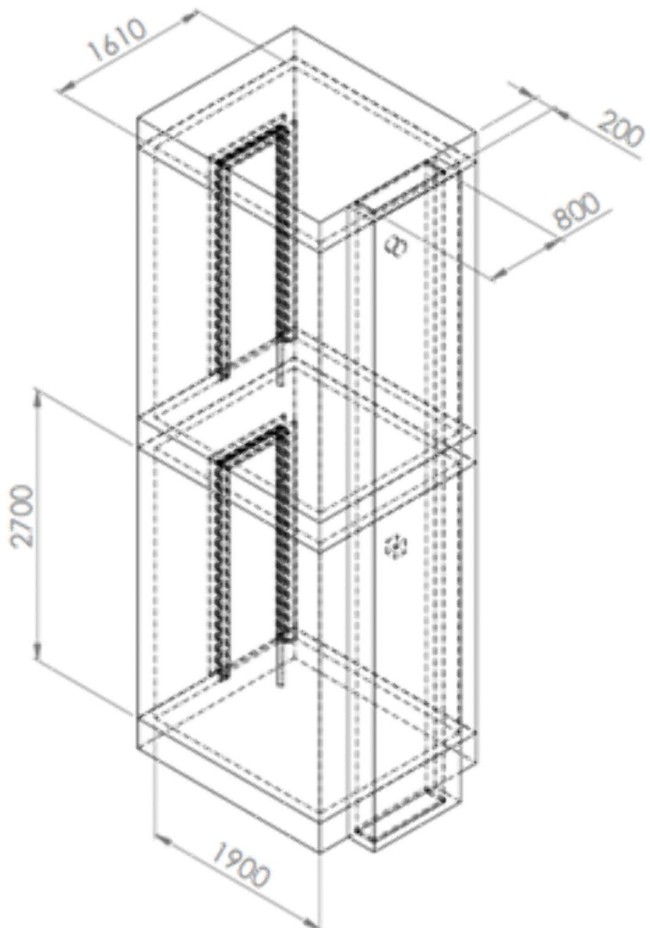

**Fig 5. Schematic representation of the CFD model geometry (dimensions in millimeters) used to model two vertically aligned, adjoining bathrooms.**

computation time without affecting air movement accuracy. This simplified CFD model includes variables such as duct roughness based on the typical roughness average of brick and mortar which is the material used for the construction of the bathroom exhaust.

The model assumed steady-state, turbulent flows of a Newtonian, incompressible fluid with constant physical properties. Simulations were conducted using the finite volume method applied to the fluid dynamics equations within the mesh generated by OpenFOAM, incorporating a conventional k-ε turbulence model and Boussinesq approximation. It was further assumed that the fluid was dry air and that no heat transfer occurred between the air and the enclosure walls, given their similar temperatures and internal location. Airflow in the bathroom duct was simulated under two conditions: with one window open to the community patio (Table 1, Case 2) and with the kitchen exhaust hood operating at maximum fan speed (Table 1, Case 3). The boundary conditions are in Table 2 for each condition.

Table 3 presents a comparison between the CFD results and measurements taken from the bathroom of the 4[th]-floor home, showing agreement between observed and predicted values. This alignment confirms that the model accurately represents airflow conditions within the homes and bathroom ventilation duct. In the simulation where the patio window is open (Figs 6 and 7), airflow exits the lower bathroom at approximately 2 m/s and enters the shared bathroom duct, rising toward the upper. A portion of this flow enters the upper bathroom through its exhaust opening at around 1 m/s. Elevated

**Table 2. Boundary Conditions for the Simulated Conditions.**

| Parameter | Patio Window Open | Kitchen Exhaust Hood On |
|---|---|---|
| Air Temperature (°C) | 20 | 20 |
| $CO_2$ Concentration in Lower Bathroom (Ppm) | 2000 | 2000 |
| Pressure at Lower Bathroom Door (Pa) | 101325 | 101325 |
| Pressure at Upper Bathroom Door (Pa) | 101299 | 101292 |
| Pressure at Upper Bathroom Duct Section (Pa) | 101300 | 101300 |
| Δp Between Bathroom Ventilation Duct and Upper Bathroom Door (Pa) | 1 | 8 |

**Table 3. Comparison of CFD Simulated Airflow Rates and Carbon Dioxide Concentrations with Measurements.**

| Parameter | Simulated | Measured |
|---|---|---|
| **Patio Window open** | | |
| Outflow in Lower Bathroom (l/s) | 62.3 | N/A |
| Inflow in Upper Bathroom (l/s) | 11.3 | 11.7 |
| $CO_2$ Concentration in Upper Bathroom (ppm) | 900 | 1032 |
| **Kitchen Exhaust Fan On** | | |
| Outflow in Lower Bathroom (l/s) | 62.4 | N/A |
| Inflow in Upper Bathroom (l/s) | 36.2 | 40 |
| $CO_2$ Concentration in Upper Bathroom (ppm) | 1550 | 1450 |

$CO_2$ concentrations are observed in the lower bathroom, with some of this $CO_2$ traveling upward through the bathroom duct and into the upper bathroom, suggesting a potential pathway for aerosol transmission. In the kitchen exhaust hood simulation (Figs 8 and 9), the same pattern occurs, but with greater intensity: airflow into the upper bathroom reaches approximately 2 m/s, and $CO_2$ levels are higher. These results indicate that operating a kitchen hood may increase the risk of aerosol transmission between bathrooms via the shared bathroom ventilation duct more than opening the patio window.

## CONTAM-Quanta modeling of transmission

A multizonal modeling approach was used to examine conditions that could lead to aerosol transmission of COVID-19 between homes in the Santander building, focusing on the shared bathroom ventilation duct (Fig 10a). Simulations were implemented using the CONTAM (version 3.4) model, developed by the U.S. National Institute of Standards and Technology (NIST), to simulate airflow and indoor infectious aerosol concentrations [15,16]. Briefly, each room was modeled as a well-mixed zone, with airflow paths defined by window openings and vertical shafts. Infiltration rates were calculated using a power-law equation that accounts for weather and pressure differences. To estimate airborne infection risk for occupants, the Wells-Riley equation was applied, relating aerosol concentrations, breathing rate (0.72 m$^3$/h), and exposure duration (assumed to be 6 hours, from 9:00–15:00). Integration of Wells-Riley with CONTAM leads to the CONTAM-Quanta Model; for full details of this modeling framework, parameter selection, and validation, refer to Yan et al. (2022) and Yan et al. (2023) [17,18]. Results from this model are expressed as the number of quanta inhaled (number of airborne particles containing virus that were inhaled and that caused an infection) and the probability of infection to evaluate occupants' risk of getting COVID-19. These methods were used to identify the factors that impact horizontal and vertical aerosol transmissions within the residential building and recommend effective mitigation strategies.

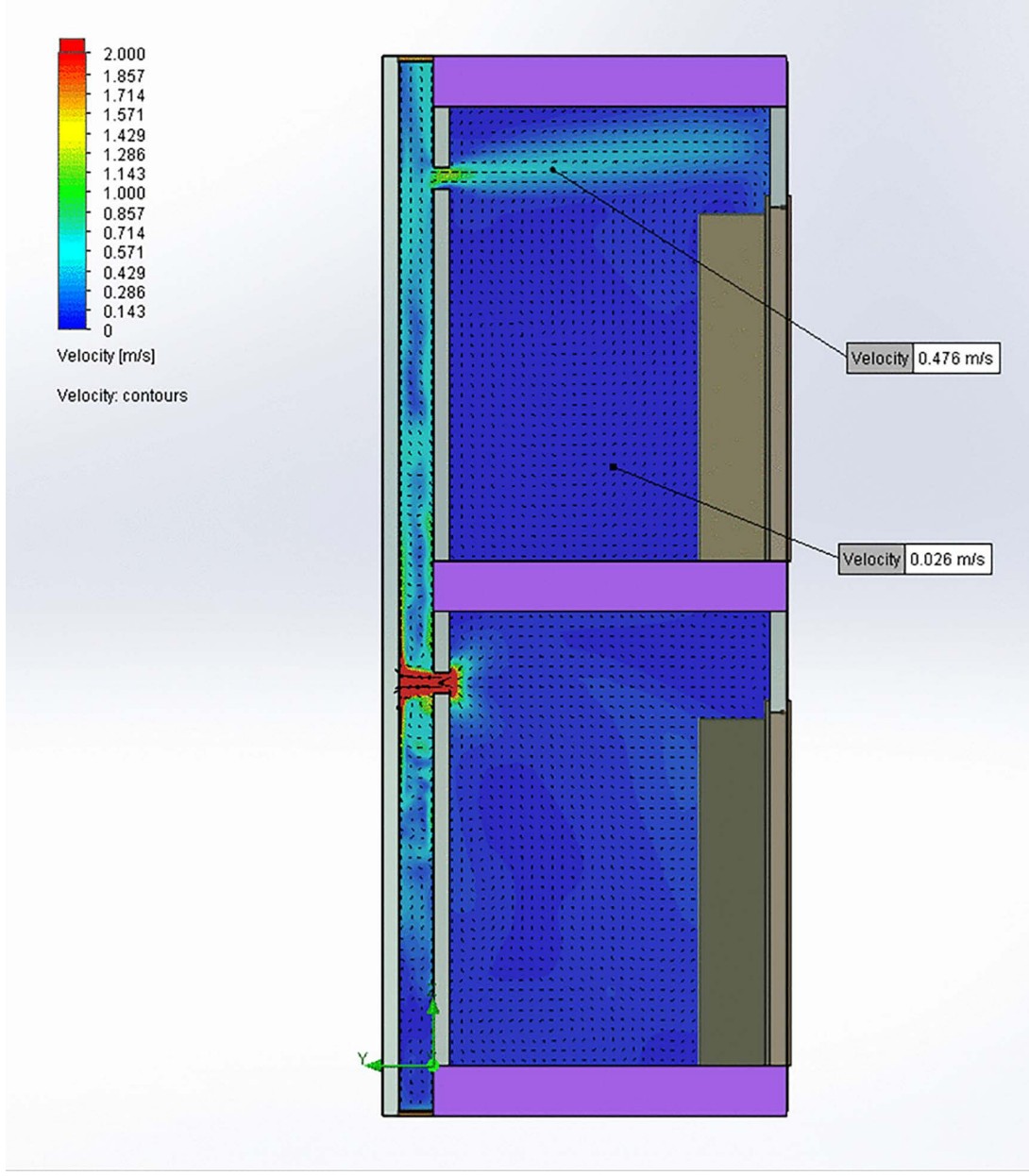

**Fig 6. Air velocities in the central plane of the bathroom ventilation duct for patio window open.**

In this study, there were 28 occupants in Homes A and B, with 6 index cases, leaving 22 susceptible individuals. If those with modified bathroom exhausts are excluded, the susceptible population is reduced to 11. To prevent any new infections among the 11 people who could be exposed, the risk for each individual must remain extremely low. To keep all 11 individuals free from infection, when they are exposed to 6 infected people, $p$ is defined as the probability that one susceptible person gets infected, and then the probability that none will get infected is $(1-p)^{11}$. Assuming this probability 95%, then is $(1-p)^{11} \geq 0.95$, and solving for $p \leq 0.0046 = 0.46\%$. If we set the probability that no one will get infected at

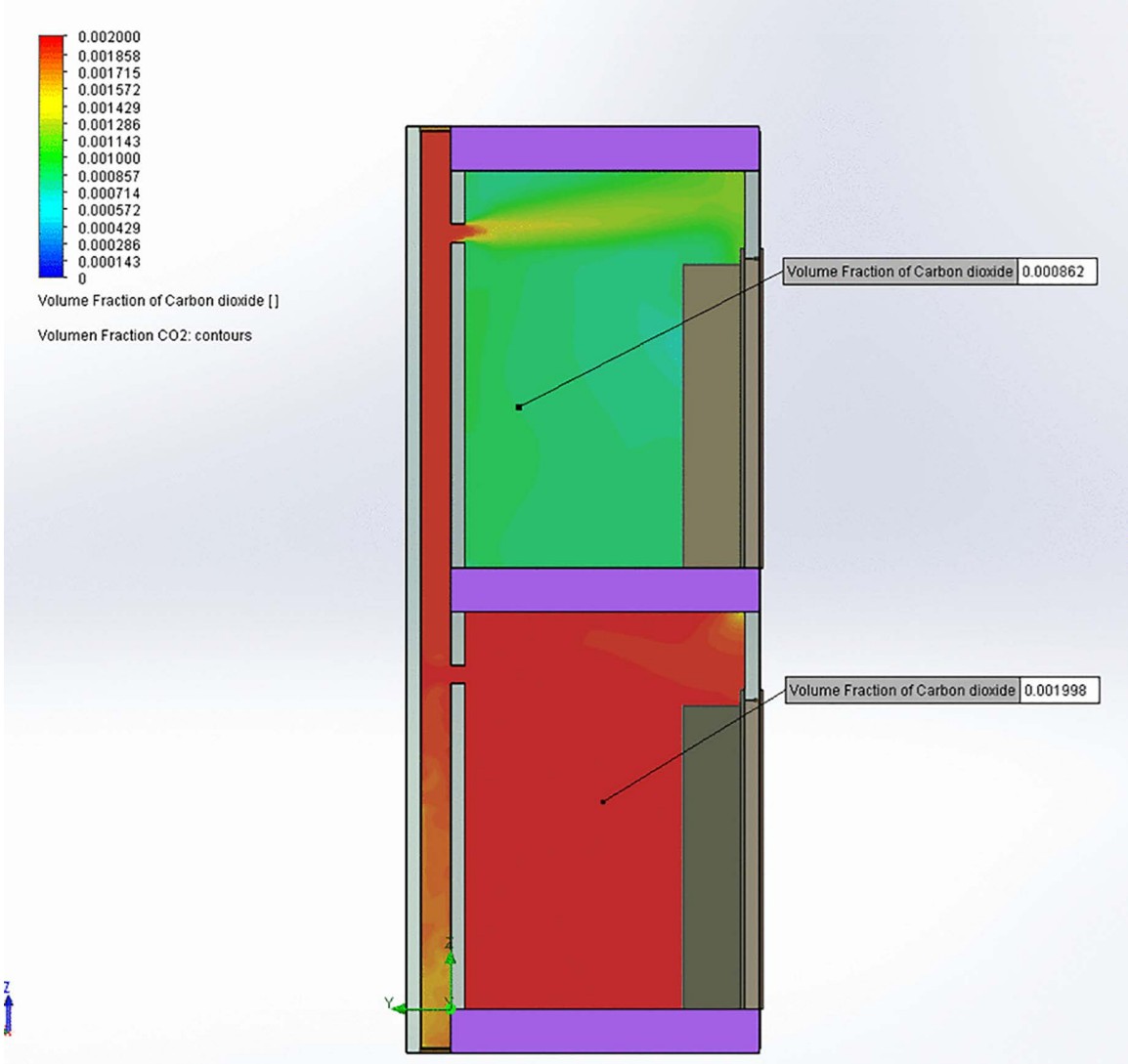

**Fig 7. CO$_2$ concentrations in the central plane of the bathroom ventilation duct for patio window open.**

99%, which is more protective for a disease like COVID-19, with such a high mortality rate initially, and in a building setting with constant exposure, then $p$ = 0.09%.

The modeled floor plan is shown in Fig 10b; the building consists of two parts, Home A and Home B, with Home A selected for investigation. The modeling focused on the possibility of transmission between the third floor, where the index cases lived, and adjacent floors. The cases investigated are presented in Table 4. In each simulation, it was assumed that on the 3rd floor of Home A, five infected individuals were in Room A, and one was in Bath A, with none of them wearing masks. A baseline case was run initially, assuming steady wind at one intensity and direction, with the window to the main street open for cooling and ventilation. Additional cases explored the influence of possible quanta transmission between the third floor and the 2nd and 3rd floors of outdoor weather (changing wind direction, wind intensity, and fluctuating weather conditions), opening interior/exterior windows, and the use of the kitchen exhaust hood and/or bathroom fan.

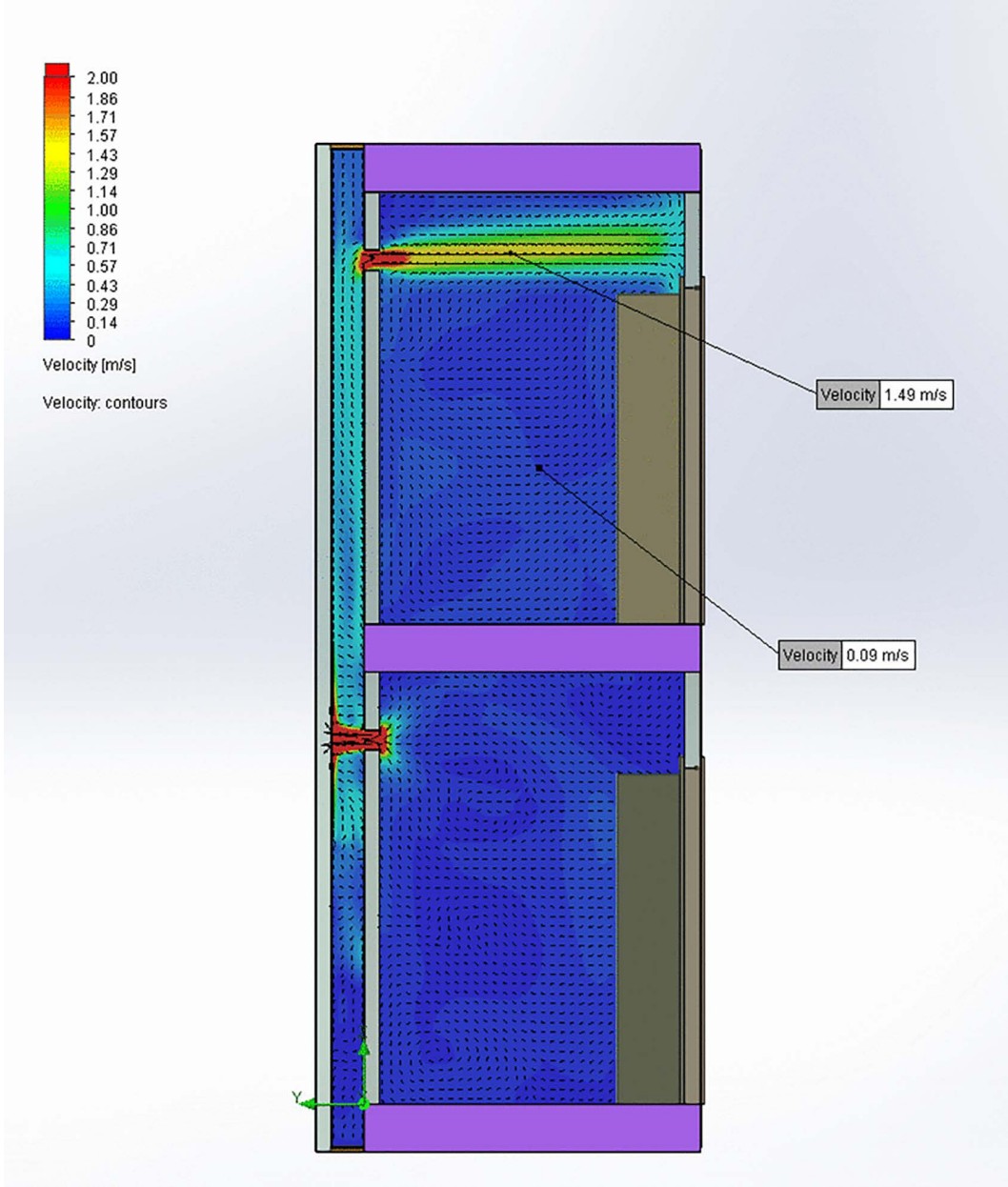

**Fig 8. Air velocities in the central plane of the bathroom ventilation duct for kitchen exhaust hood on.**

Airflow and pressurization measurements from the 4th-floor home of the Santander building (Table 1) verified that the building surface area and leakage rates assumed in the CONTAM-Quanta Model were reasonable. The modeled bathroom airflow rate on July 7, 2022, from 9:00–15:00 was compared with measurements. Fig 11 shows a good match, with the coefficient of determination $r^2 = 0.83$.

Figs 12 and 13 illustrate quanta concentration patterns on the 2nd, 3rd, and 4th floors under differing wind, weather, and window conditions (Cases 1–6, Table 1). White areas indicate regions free from quanta, while the colored areas represent concentrations, ranging from low (blue) to high (red). In all cases, the highest concentrations occurred in the source zones

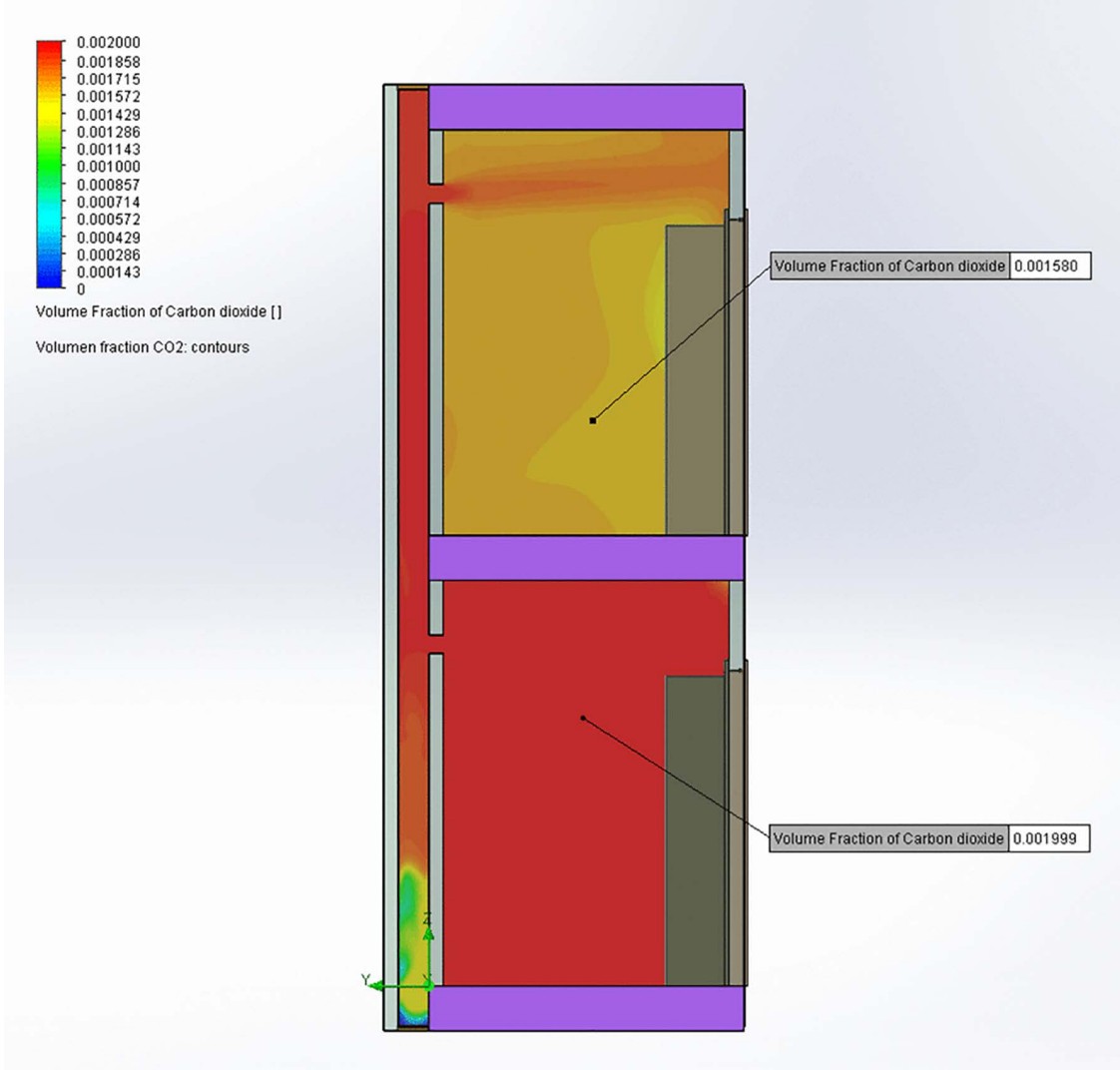

**Fig 9. $CO_2$ concentrations in the central plane of the bathroom ventilation duct for kitchen exhaust hood on.**

(Room A and Bath A), where the index cases were located. Elevated levels were also observed in adjacent areas such as Corridor A and Hall C.

Comparing Case 1 (East-West wind) and Case 2 (West-East wind) demonstrates that opening the 4th-floor window allowed wind direction to aid in lowering overall quanta concentrations. When the window to the main street is closed on the 4th floor (Case 4), Home A on that level showed increased quanta due to reduced natural ventilation. The patio had elevated quanta levels on both the 3rd and 4th floors for all cases, suggesting some degree of inter-zone transport.

The highest patio concentrations were in Case 5, which uses Santander's July weather in the simulation, the same period as the outbreak. Simulating the actual July weather patterns (Case 6) also led to increased quanta in the Home A corridor on the 4th floor. These results suggest that weather conditions impact interzonal transport when a building relies on natural ventilation and the stack effect to remove infectious aerosol. Also, it is important to use actual weather conditions where possible in CONTAM to obtain precise results, that opening windows is needed to reduce quanta concentrations, and that the patio of the building is a source of infectious aerosol.

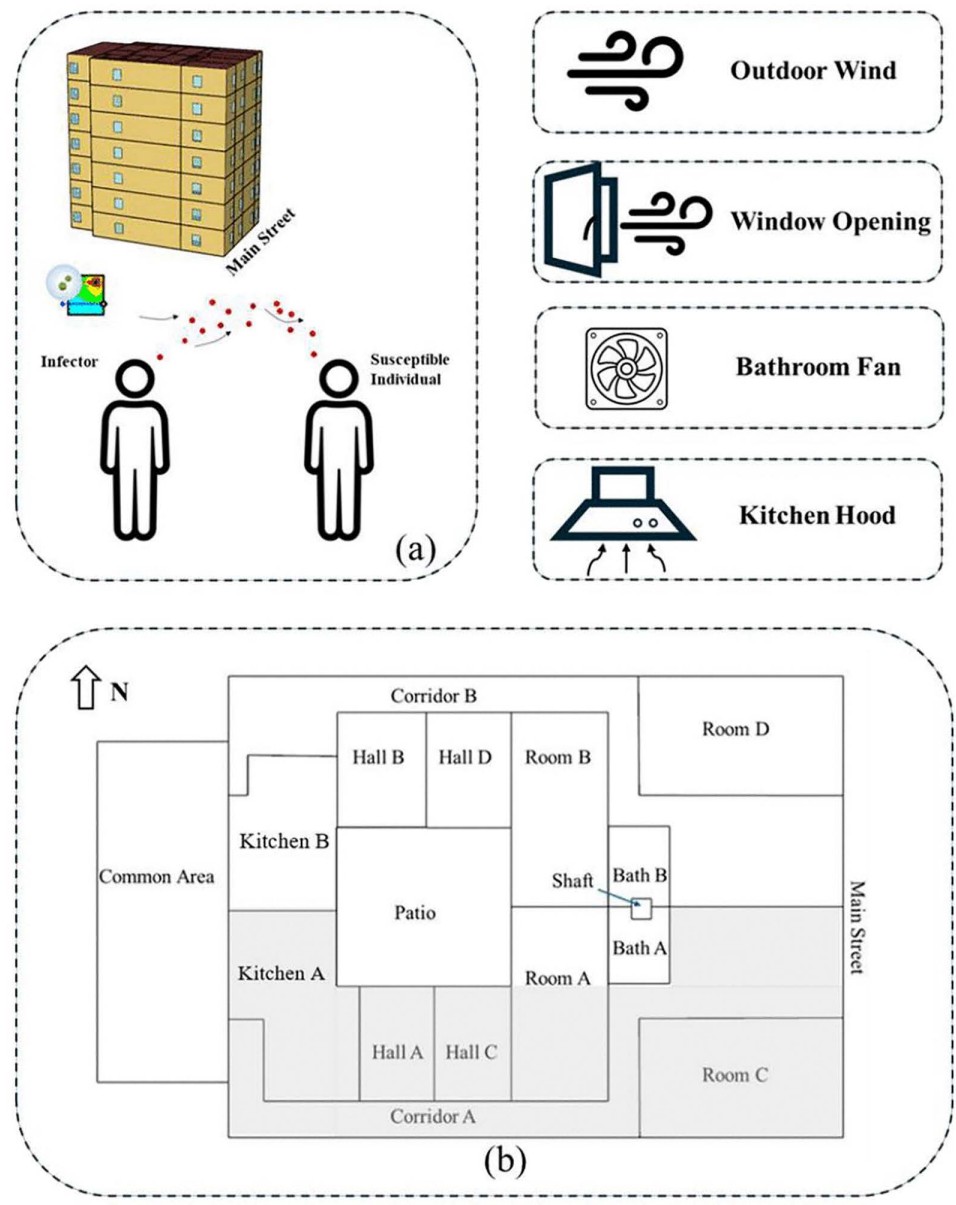

**Fig 10. CONTAM-Quanta model for the Santander residential building, Northern Spain.** The building is shown in (a), and the floor plan of each floor is outlined in (b), with the gray area indicating Home A, which is where the infected residents lived on the 3rd floor. The index cases were located on the 3rd floor, specifically in Home A (Rooms A and Bath A). Simulations were conducted exploring the impacts of outdoor wind, window opening, and bathroom fan and kitchen operation on airborne infection.

**Table 4. Cases Investigated with the CONTAM-Quanta Model\*.**

| Case | Wind direction | Wind degree | Wind intensity | Patio window | Main street window | Bathroom fan | Extract hood |
|------|---------------|-------------|----------------|--------------|-------------------|--------------|--------------|
| Case 1 (Baseline) | East-West | 270 | 1.39 m/s | All closed | 4th floor open | Off | Off |
| Case 2 | *West-East* | *90* | 1.39 m/s | All closed | 4th floor open | Off | Off |
| Case 3 | East-West | 270 | *2.78 m/s* | All closed | 4th floor open | Off | Off |
| Case 4 | East-West | 270 | 1.39 m/s | All closed | *All closed* | Off | Off |
| Case 5 | *Weather 7/7/22* | -- | -- | All closed | 4th floor open | Off | Off |
| Case 6 | *Weather 7/7/22* | -- | -- | All closed | *All closed* | Off | Off |
| Case 7 | *Weather 7/7/22* | -- | -- | *3rd & 4th floor open* | 4th floor open | Off | Off |
| Case 8 | *Weather 7/7/22* | -- | -- | *3rd & 4th floor open* | *All closed* | Off | Off |
| Case 9 | *Weather 7/7/22* | -- | -- | All closed | *3rd & 4th floor open* | Off | Off |
| Case 10 | *Weather 7/7/22* | -- | -- | All closed | *All closed* | *On – 3rd floor* | Off |
| Case 11 | *Weather 7/7/22* | -- | -- | All closed | *All closed* | *On – 3rd & 4th floor* | Off |
| Case 12 | *Weather 7/7/22* | -- | -- | All closed | 4th floor open | *On – 3rd & 4th floor* | Off |
| Case 13 | *Weather 7/7/22* | -- | -- | All closed | *All closed* | Off | *On – 4th floor* |
| Case 14 | *Weather 7/7/22* | -- | -- | All closed | *All closed* | *On – 3rd floor* | *On – 4th floor* |
| Case 15 | *Weather 7/7/22* | -- | -- | All closed | 4th floor open | *On – 3rd floor* | *On – 4th floor* |

\***Bold-italic** indicates a change in the parameter relative to the Baseline Case.

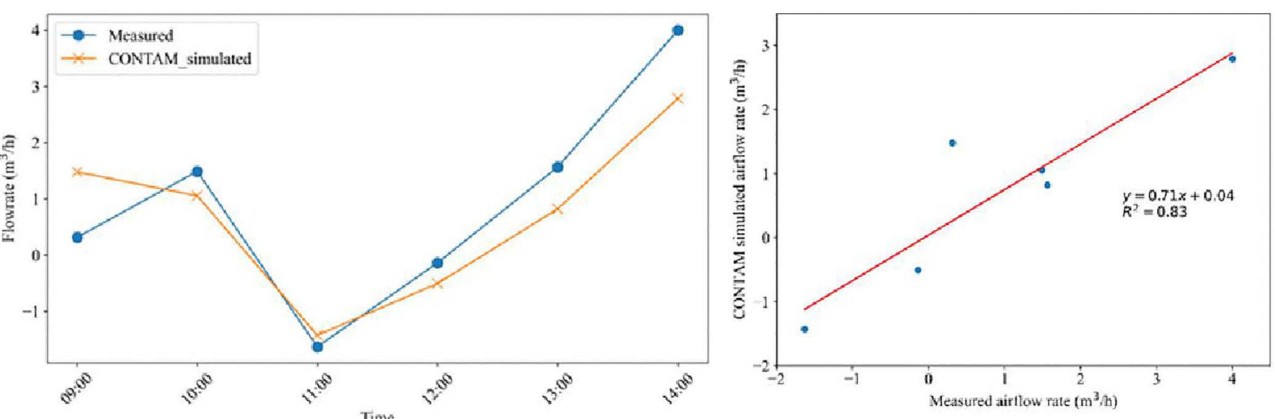

**Fig 11. Comparison between CONTAM Model simulated airflows and 4th Floor bathroom Santander building field measurements.** The flow rate time profile is shown in (a), and the correlation between model and measurements is in (b).

In Fig 14, the probability that one susceptible person gets infected $p$, is illustrated. The highest infection risk occurs in the source zones, while adjacent rooms on the same floor of Home A experience relatively low infection risk. The 2nd and 4th floors have low quanta concentrations, which translate into low infection risk across cases 1–6; thus, the investigated changes in wind and weather did not increase the transmission risk.

Opening the windows on the 3rd and 4th floors of the building introduces additional airflow pathways. When the windows to the patio are open in Room A, aerosols migrate upward to the 4th floor, resulting in elevated quanta concentrations in Room A, Hall A, and Hall C (Fig 15, Cases 7 and 8). Conversely, keeping the patio windows closed while opening the street-facing windows (Case 9) promotes dilution of aerosols by outdoor air, limiting interzonal transport. Fig 16 shows

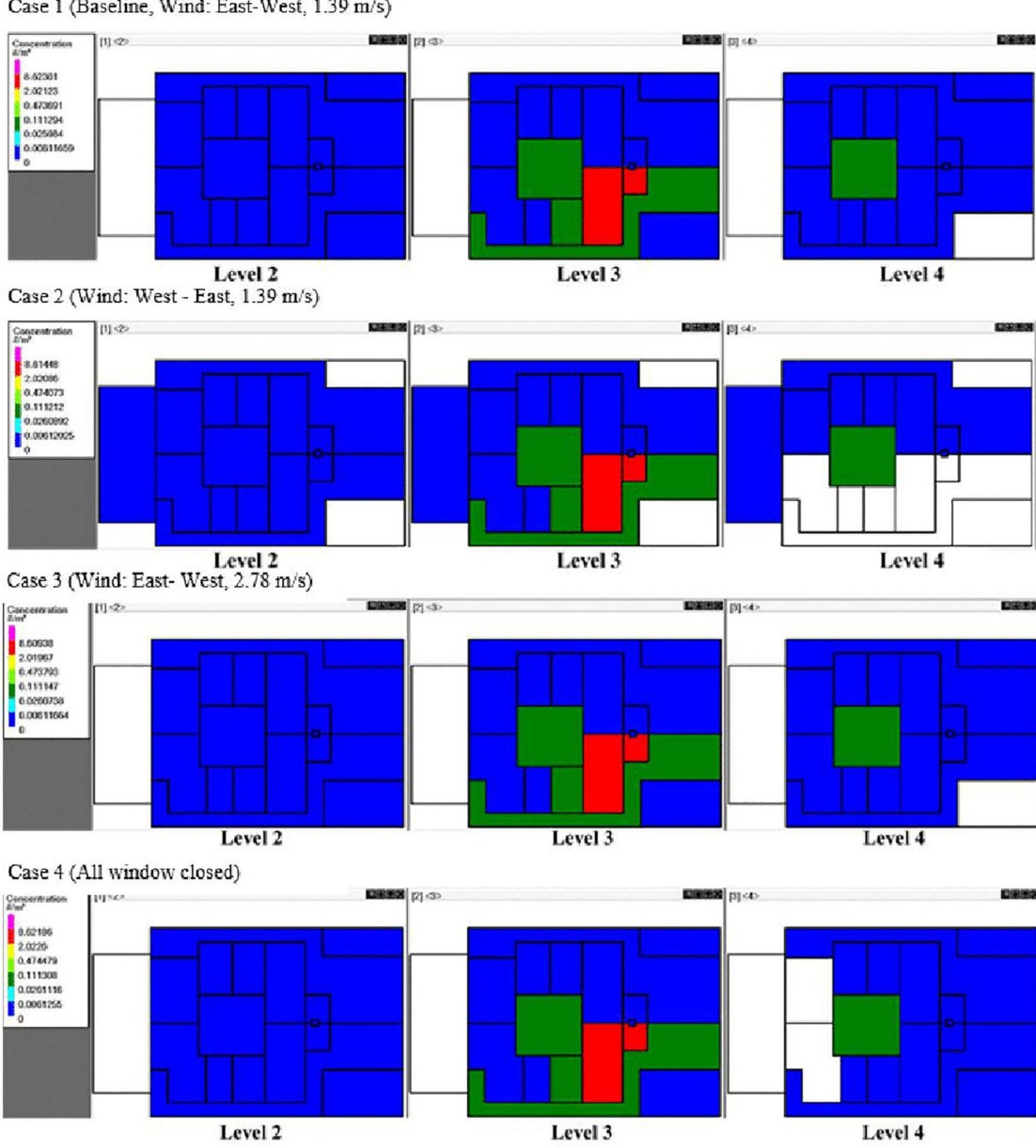

**Fig 12. Schematics showing quanta concentrations for CONTAM-Quanta simulation Cases 1–4 on different floors of the building and in different rooms.** The index cases were located on the 3rd floor, Home A. Five infected individuals were in Room A, and one infected individual was in Bath A on level 3. The white color indicates zero quanta concentration, the dark green color is 0.03-0.11 quanta/m³, and the red color is 2.0-8.6 quanta/m³.

that opening the patio windows connected to Room A on the 3rd and 4th floors can lead to a small but noticeable infection risk on the 4th floor, as aerosols migrate vertically through the patio shaft. But there is no additional risk on the 4th floor when the street-facing windows are open. These results indicated that wind entering through street-facing windows was effective in diluting quanta and mitigating transmission risk on both floors.

Bathrooms usually represent high-risk environments for airborne infection transmission due to activities such as toilet flushing, which can generate pathogen-laden aerosols with persistent suspension in confined spaces. Mechanical

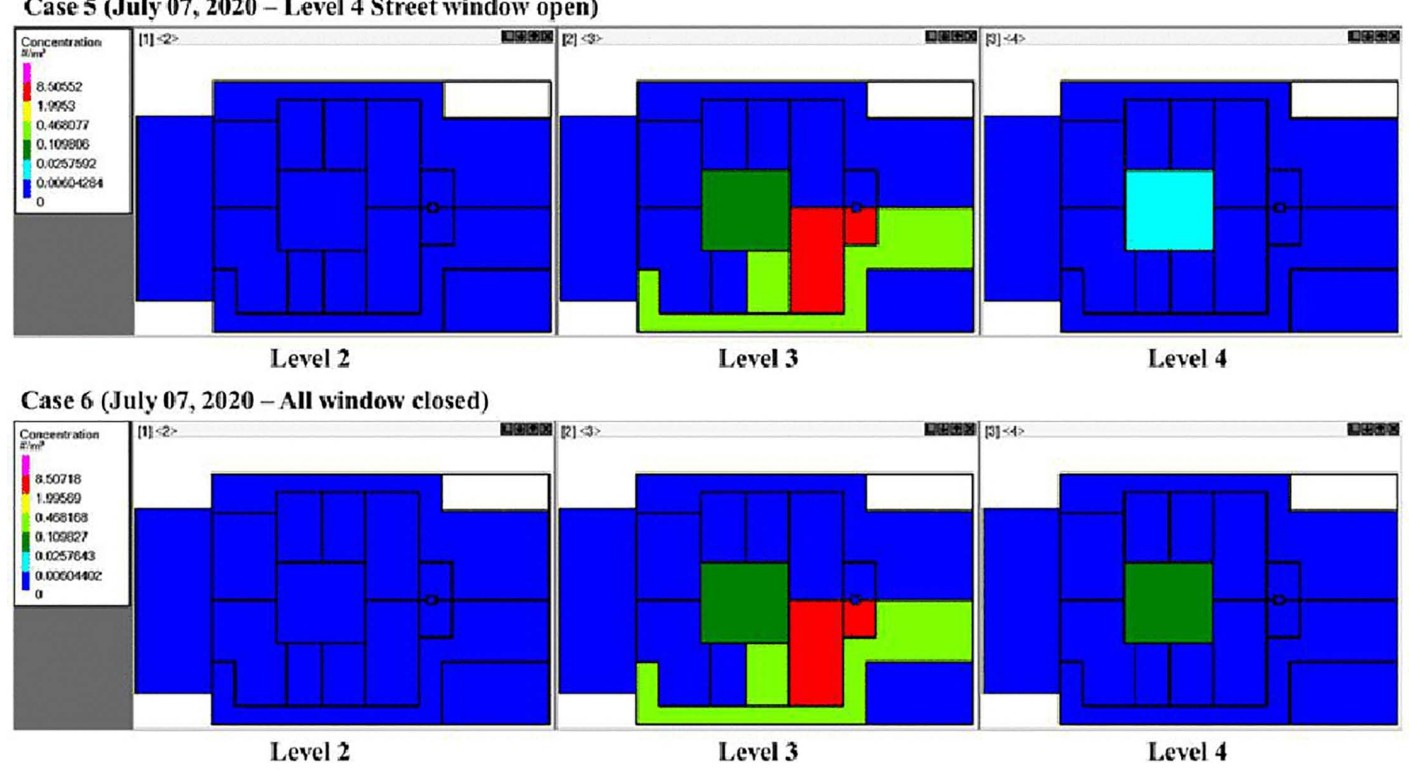

**Fig 13. Schematics showing quanta concentrations for CONTAM-Quanta simulation Cases 5 and 6 on different floors of the building and in different rooms.** The white color indicates zero quanta concentration, the dark green color is 0.03-0.11 quanta/m³, and the red color is 2.0-8.5 quanta/m³. Five infected individuals were in Room A, and one infected individual was in Bath A, on level 3.

ventilation systems, such as bathroom exhaust fans, are often installed to regulate humidity and odors; however, their operation inherently alters indoor pressure differentials and airflow trajectories, which are critical factors governing aerosol transport. Bathroom fan simulation results (Cases 10–12) are shown in Fig 17. The fan operation promoted quanta transport to the 2nd floor by drawing quanta into the shared vertical bathroom ventilation duct that connects Baths A and B. Once these quanta reached the 2nd floor, they spread to adjacent zones: Room A, Corridor A, Room B, and Corridor B. This pattern was observed across all three cases, suggesting that running bathroom fans inadvertently enabled cross-floor contamination via interconnected shafts. The operation of the bathroom fans resulted in increased infection risk on floor 2 bathrooms, due to increased quanta in the vertical bathroom ventilation duct (Fig 18). In Bath A, the risk was found to be greater than 0.09% but remained below 0.46%, within the threshold for additional transmission. Infection risk is not increased on floor 4, possibly due to the street window being open (Case 12).

In residential buildings, kitchen exhaust hoods are essential for maintaining indoor air quality by removing cooking-related pollutants through the creation of localized negative pressure zones. However, their operation can unintentionally disrupt indoor airflow patterns. When exhaust rates are high and makeup air is insufficient, the resulting depressurization may draw air and potentially contaminants from adjacent areas, including bathrooms or other infectious zones. This study examined three scenarios involving kitchen-hood operation. In Case 13 (Fig 19), activating the kitchen hood on the 4th floor increased the movement of quanta to that floor, while limiting horizontal transport on the 3rd floor. When the bathroom fan was simultaneously used on the third floor (Case 14), quanta concentrations rose on the 4th floor, indicating enhanced vertical transmission. Opening the street-facing window on the 4th floor (Case 15) facilitated the dilution and removal of

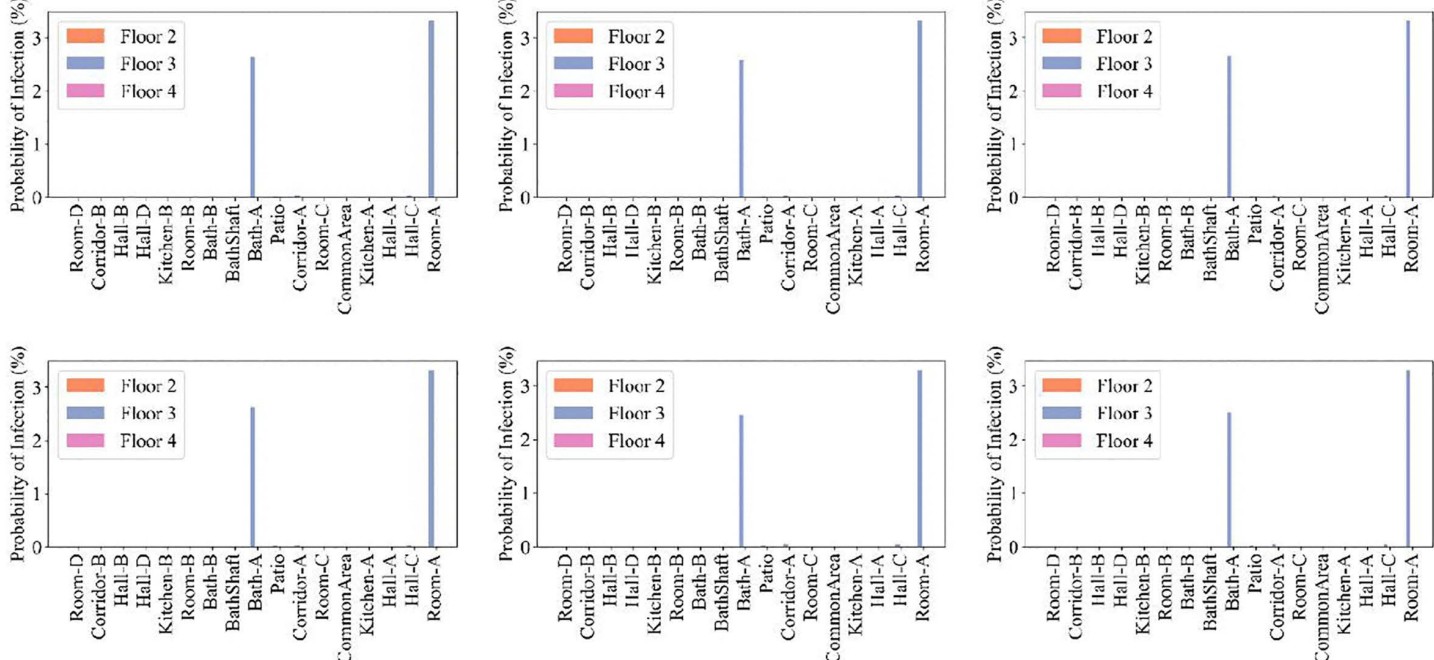

**Fig 14. Probability of infection on the 2ⁿᵈ, 3ʳᵈ, and 4ᵗʰ floors for every room that was simulated in the CONTAM model.** Panels (a) through (f) correspond to wind and weather simulation Cases 1 through 6. Five infected individuals were in Room A, and one infected individual was in Bath A, on the 3ʳᵈ floor.

quanta near the window, improving safety in that area. However, this also altered the airflow within the vertical bathroom ventilation duct, resulting in increased transport to the 2ⁿᵈ floor. As shown in Fig 20, infection probability increased on the 4ᵗʰ floor during kitchen-hood operation, and on the 2ⁿᵈ floor when the bathroom exhaust fan was running on the 3ʳᵈ floor. As Case 15 demonstrates, the operation of the kitchen hood on the 4ᵗʰ floor extracted quanta from the 3ʳᵈ floor index zones; however, when the bathroom fan on the 3ʳᵈ floor was turned on, this effect was mitigated, but more quanta were transmitted to the 2ⁿᵈ floor.

## Conclusions

Based on epidemiological data, field observations, genetic sequencing, and modeling studies, the vertical bathroom ventilation duct within the homes was identified as the likely pathway of contagion. Air transferred through shared bathroom ducts exposed residents of four homes to SARS-CoV-2, despite low community transmission in the city. Homes with covered exhaust grills or individual extractors were not affected, nor were other homes without shared bathroom ducts. During the comprehensive screening of the building and neighboring buildings for COVID-19 cases during the original outbreak, it was determined that no other residences or individuals from adjacent vertical shafts were affected. This assessment included visitors to the building and those in close contact.

The results of this study reinforce the conclusion that the operation of kitchen hoods, bathroom fans, or the opening of street-facing windows can substantially affect and introduce variability in infectious aerosol transport and airborne infection transmission patterns. In real life, the daily activities of residents are random, and multiple behaviors may occur simultaneously, increasing the unpredictability of aerosol movement in residential buildings due to zonal airflows. These impacts can sometimes contradict one another. Consequently, residents' aerosol exposure risks can be substantially affected by

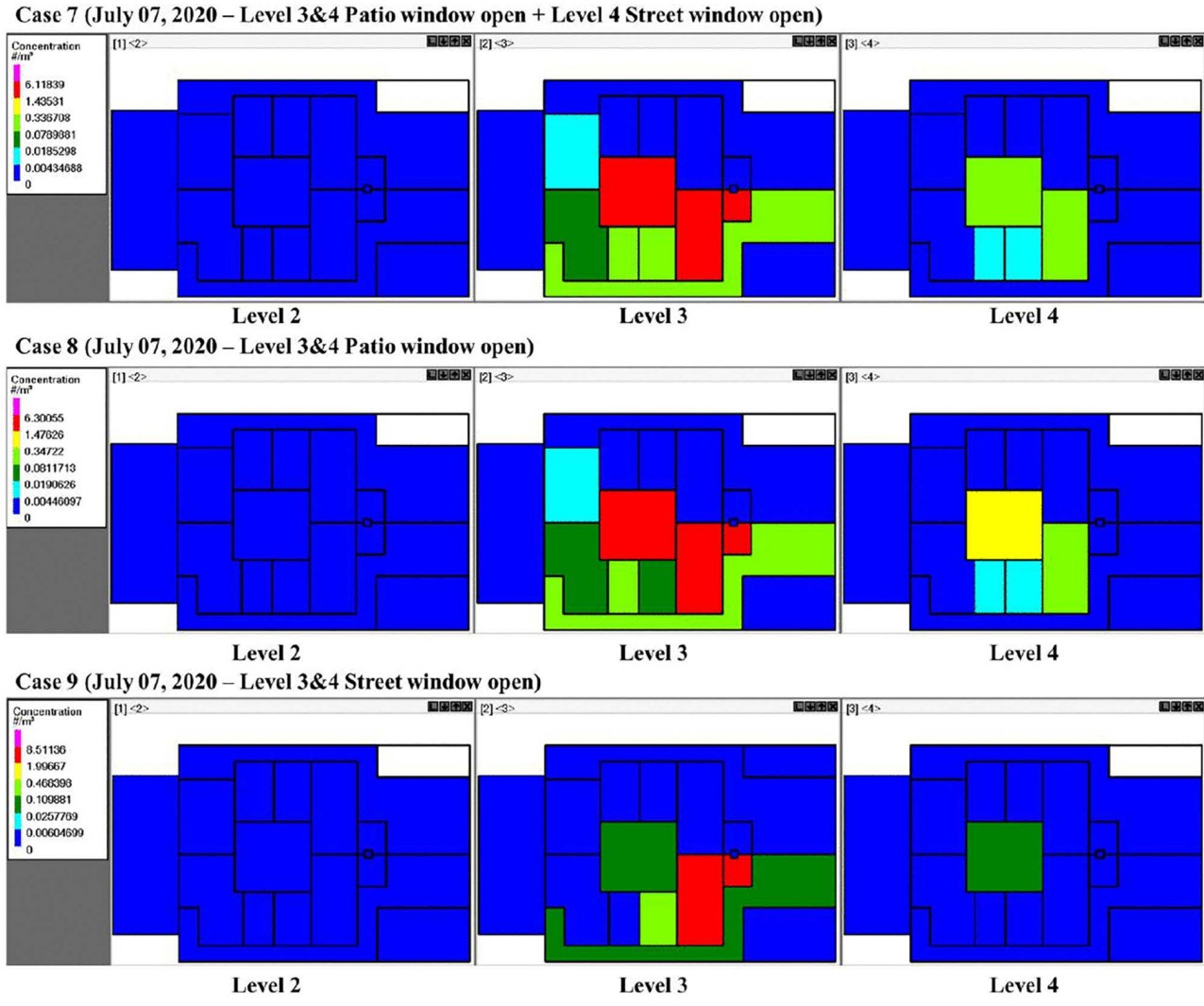

**Fig 15. Schematics showing quanta concentrations for CONTAM simulation Cases 7-9 on different floors of the building and in different rooms.** The white color indicates zero quanta concentration, the dark green color is 0.03-0.11 quanta/m³, and the red color is 2.0-8.5 quanta/m³. Five infected individuals were in Room A, and one infected individual was in Bath A, on level 3.

the activities of their neighbors, leading to considerable uncertainty in infection transmission during outbreaks. It should be noted that the relative magnitudes of these impacts were not compared in this study but may be addressed and investigated in future work.

This study employed measurements and modeling to investigate how infectious aerosols could be transported within and between homes in the Santander multi-floor residential building. Results show that relying on natural convection, or the stack effect, in bathroom ventilation ducts does not prevent contaminants from being transported between homes. Specifically, susceptible individuals in poorly ventilated indoor spaces may be exposed to airborne respiratory pathogens through connected ventilation systems. This exposure can result in a sufficient viral load to transmit the disease between homes, even in the absence of direct physical contact.

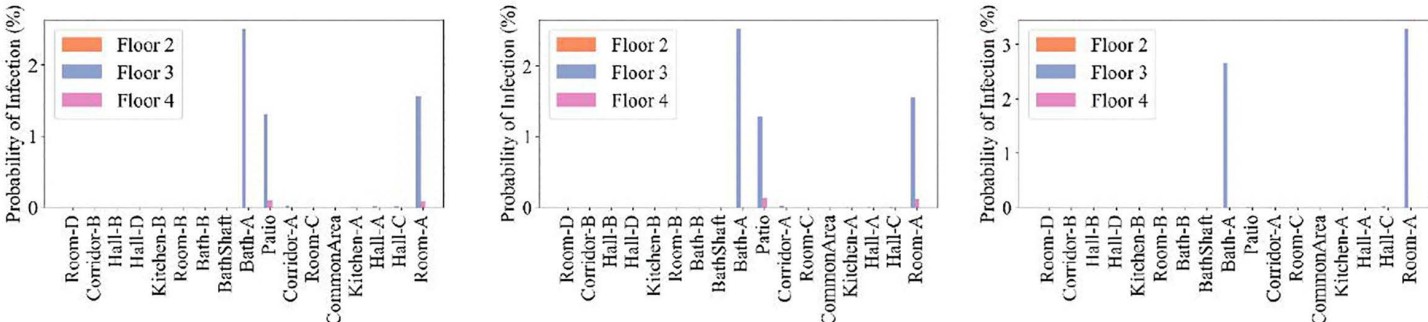

**Fig 16. Probability of infection on the 2nd, 3rd, and 4th floors. Panels (a) through (c) correspond to window opening Cases 7 through 9.** Five infected individuals were in Room A, and one infected individual was in Bath A, on the 3rd floor.

**Case 10 (July 07, 2020 – Level 3 Bathroom fan operating at 28.8 m³/h)**

Level 2   Level 3   Level 4

**Case 11 (July 07, 2020 – Level 3 &4 Bathroom fan operating at 28.8 m³/h)**

Level 2   Level 3   Level 4

**Case 12 (July 07, 2020 – Level 3 &4 Bathroom fan operating at 28.8 m³/h, Level4 Street window open)**

Level 2   Level 3   Level 4

**Fig 17. Schematics showing quanta concentrations for CONTAM simulation Cases 10–12 on different floors of the building and in different rooms.** The white color indicates zero quanta concentration, the dark green color is 0.03-0.11 quanta/m3, and the red color is 2.0-8.5 quanta/m3. Five infected individuals were in Room A, and one infected individual was in Bath A, on level 3.

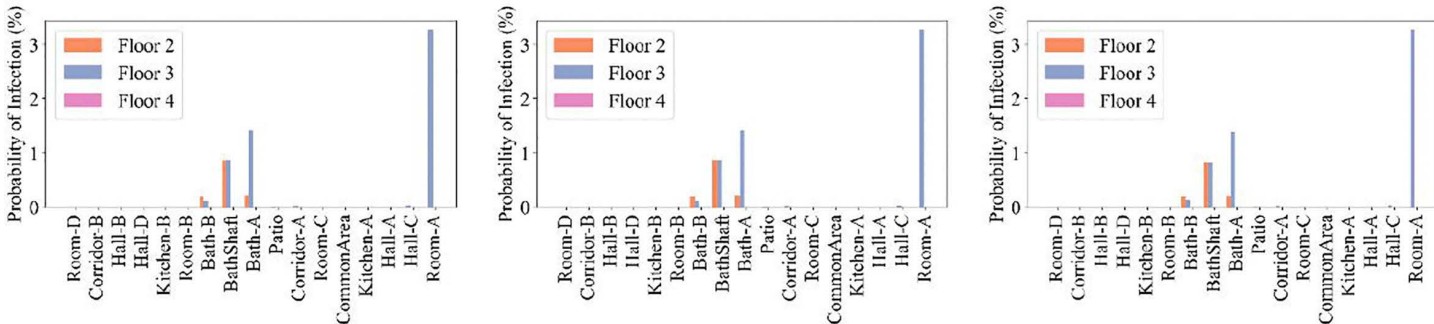

**Fig 18. Probability of infection on the 2ⁿᵈ, 3ʳᵈ, and 4ᵗʰ floors. Panels (a) through (c) correspond to Bathroom fan simulations Cases 10 through 12.** Five infected individuals were in Room A, and one infected individual was in Bath A, on the 3ʳᵈ floor.

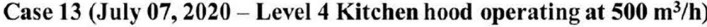

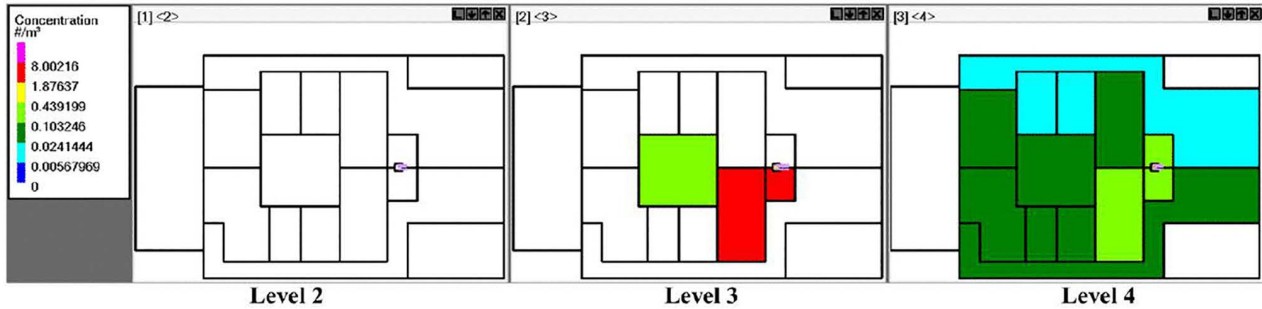

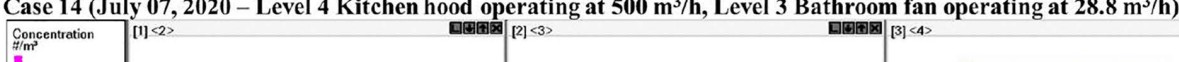

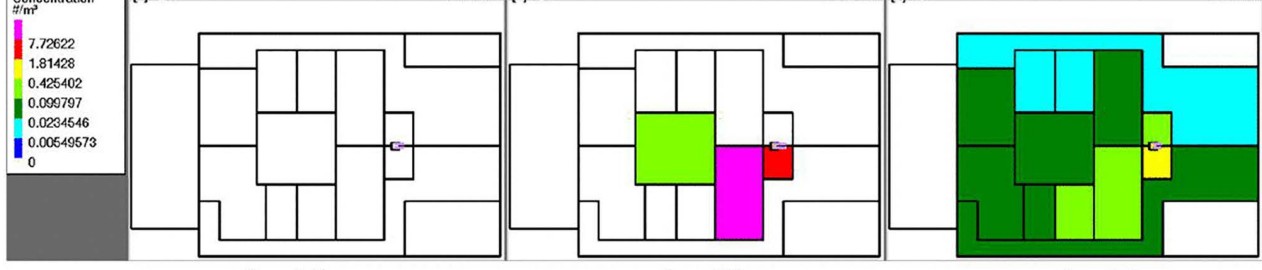

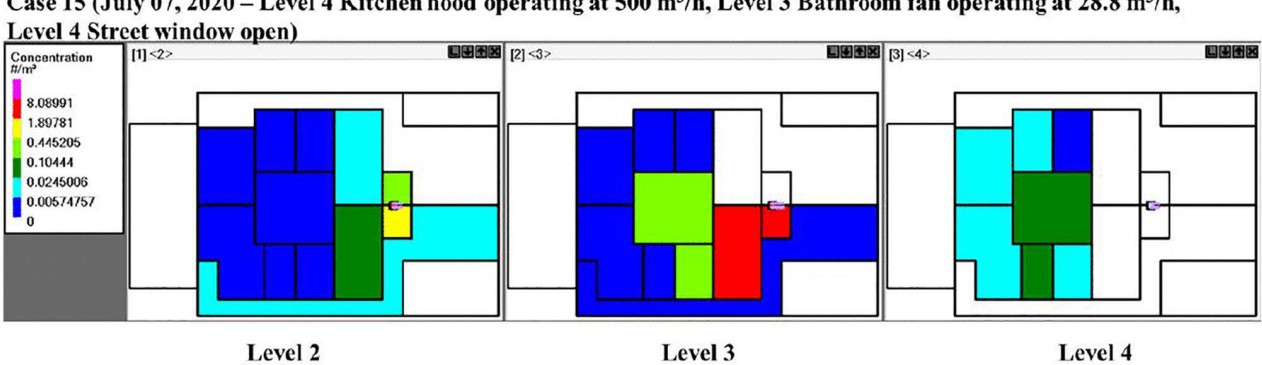

**Fig 19. Schematics showing quanta concentrations for CONTAM simulation Cases 13–15 on different floors of the building and in different rooms.** The white color indicates zero quanta concentration, the dark green color is 0.02-0.10 quanta/m³, and the red color is 1.8-8.0 quanta/m³. Five infected individuals were in Room A, and one infected individual was in Bath A on level 3.

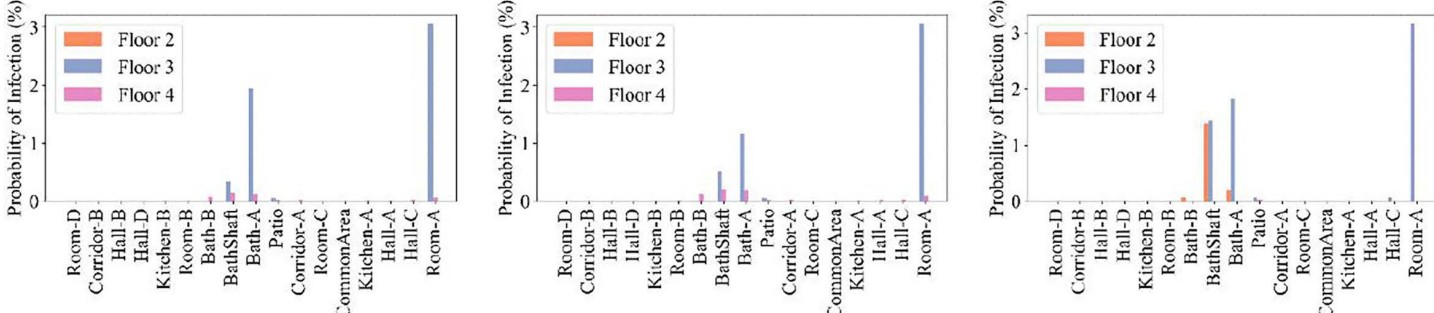

**Fig 20. Probability of infection on the 2ⁿᵈ, 3ʳᵈ, and 4th floors.** Panels (a) through (c) correspond to kitchen exhaust Cases 13 through 15. Five infected individuals were in Room A, and one infected individual was in Bath A, on the 3ʳᵈ floor.

The CONTAM-Quanta model was used to predict the probability of an additional infection under different airflow conditions due to weather, wind, window opening, or exhaust fan operation. Infection risk was always elevated on the third floor in the bathroom A and Room A, where the index cases were assumed to be present, above the threshold risk of 0.09% needed to add additional infections. Risk was increased slightly on the 2ⁿᵈ and 4ᵗʰ floors for some scenarios, indicating that transmission risk is not negligible. Other scenarios may introduce elevated risk of infection; however, the number of scenarios that could be investigated had to be limited to a reasonable number of simulations. These results show what is possible and that interzonal transport can occur, resulting in infection risk.

An effective engineering control solution to reduce the potential for contamination between homes is the installation of forced air exhaust in the bathrooms by placing an extraction fan in the exhaust opening. These extraction fans can be equipped with a flap to prevent air from flowing into the home when the fan is not being used. These fans are typically inexpensive and represent a great cost-benefit solution to new constructions or retrofitting into existing building structures. Additionally, in buildings with shared ducts, sufficient make up air is needed to prevent kitchen exhaust use from drawing air—potentially containing pathogens—from neighboring apartments. This can be achieved by opening a street-facing window.

This investigation highlights the importance of integrating scientific knowledge about airborne transmission into building inspection protocols and the broader management of infectious diseases within built environments. Early detection of outbreaks should prioritize identifying clusters of cases that share common building elements—such as ventilation ducts, drainage pipes, or architectural cavities—that can facilitate the movement of airborne or chemical contaminants between homes. The evidence and engineering solutions presented here underscore the need for proactive assessment and targeted interventions, ensuring that building design and maintenance practices evolve to mitigate future risks. By recognizing the role of shared infrastructure in disease propagation, authorities and building managers can implement timely measures to protect occupant health and prevent the spread of airborne pathogens.

## Supporting information

**S1 File. Background and Preparatory Steps for Investigating the COVID-19 Outbreak Related to Bathroom Ventilation Exhaust Ducts.**
(DOCX)

**S2 File. Building Permits Cover Letter.**
(DOCX)

**S3 File. Autorización Investigación COVID.**
(DOCX)

**S4 File. Calle Nicolas Salmeron 4 Letter.**
(DOCX)

## Acknowledgments

We are grateful to the following individuals for inspiration and support: Margarita del Val, Lidia Morawska, Xavier Querol, and Jose L. Jiménez Palacios.

## Author contributions

**Conceptualization:** Shelly L. Miller, Alberto Garcia, Liangzhu Leon Wang, Zhiqiang Zhai, Jose Ramon Aranda, Ignacio Lombillo, Delfín Silió, L. David Higuera.

**Data curation:** Shujie Yan, Jose Ramon Aranda, Fernando González-Candelas, Ernesto Cabrillo, L. David Higuera.

**Formal analysis:** Shujie Yan, Jose Ramon Aranda, Ignacio Lombillo, Javier Balbás, Delfín Silió, L. David Higuera.

**Investigation:** Shelly L. Miller, Shujie Yan, Alberto Garcia, Zhiqiang Zhai, Jose Ramon Aranda, Fernando González-Candelas, Ignacio Lombillo, Ernesto Cabrillo, Delfín Silió, L. David Higuera.

**Methodology:** Shelly L. Miller, Shujie Yan, Alberto Garcia, Liangzhu Leon Wang, Zhiqiang Zhai, Jose Ramon Aranda, L. David Higuera.

**Project administration:** Shelly L. Miller.

**Resources:** Shelly L. Miller, Shujie Yan, Jose Ramon Aranda.

**Software:** Shujie Yan, Jose Ramon Aranda.

**Supervision:** Shelly L. Miller, Liangzhu Leon Wang.

**Validation:** Shelly L. Miller, Shujie Yan, Jose Ramon Aranda.

**Visualization:** Shelly L. Miller, Shujie Yan, Jose Ramon Aranda, L. David Higuera.

**Writing – original draft:** Shelly L. Miller, Shujie Yan, Alberto Garcia, Jose Ramon Aranda, L. David Higuera.

**Writing – review & editing:** Shelly L. Miller, Shujie Yan, Alberto Garcia, Liangzhu Leon Wang, Zhiqiang Zhai, Jose Ramon Aranda, L. David Higuera.

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
