## [Decision Letter · Decision Letter 0]

2 Jan 2026

PONE-D-25-62493Potential Airborne Transmission of SARS-CoV-2 Through Bathroom Ventilation Ducts Associated with an Outbreak in a Residential Building in Santander, Spain, 2020PLOS One

Dear Dr. Miller,

Thank you for submitting your manuscript to PLOS ONE. After careful consideration, we feel that it has merit but does not fully meet PLOS ONE’s publication criteria as it currently stands. Therefore, we invite you to submit a revised version of the manuscript that addresses the points raised during the review process.

We look forward to receiving your revised manuscript.

Kind regards,

Baohua Wen

Academic Editor

PLOS One

Journal Requirements:

4. Please upload copy of Figures 7, 8, 9, 10, and 11 to which you refer in your text on pages 14, 15, 17 and 18. If the figures are no longer to be included as part of the submission please remove all reference to it within the text.

6. Thank you for providing your underlying data as Supporting Information.

We note that the data set contains text or data that is not in English. Please note that PLOS is an English-language publisher, so we require data sets to be provided in English as well. Please upload an English-language version of your data set.

This will also allow us to determine if your data follows PLOS standards per our Data Availability policy here: https://journals.plos.org/plosone/s/data-availability

Reviewers' comments:

Reviewer's Responses to Questions

**Comments to the Author**

1. Is the manuscript technically sound, and do the data support the conclusions?

Reviewer #1: Yes

Reviewer #2: Yes

2. Has the statistical analysis been performed appropriately and rigorously? 

Reviewer #1: I Don't Know

Reviewer #2: N/A

3. Have the authors made all data underlying the findings in their manuscript fully available?

Reviewer #1: Yes

Reviewer #2: Yes

4. Is the manuscript presented in an intelligible fashion and written in standard English?

Reviewer #1: Yes

Reviewer #2: Yes

5. Review Comments to the Author

Reviewer #1: The manuscript presents a compelling multidisciplinary investigation into potential airborne SARS-CoV-2 transmission via shared bathroom ventilation ducts in a residential building. The integration of epidemiological data, genetic sequencing, field measurements, CFD simulations, and multi-zone modeling provides robust evidence for the proposed transmission route. Strengths include rigorous model validation (e.g., r² = 0.83 for airflow comparisons) and practical recommendations for mitigation, such as installing exhaust fans with non-return flaps.

Minor suggestions:

Expand on the limitations of accessing only one apartment for measurements; discuss how this might affect generalizability to other floors or buildings.

In the discussion of quanta concentrations (e.g., Figs. 12-13), consider adding quantitative summaries (e.g., mean or range values) for key zones to complement the visual patterns.

Overall, the work is technically sound and contributes valuable insights to indoor aerosol transmission in multifamily housing. I recommend acceptance after minor revisions.

Reviewer #2: 1. Executive Summary

This manuscript investigates a vertically clustered outbreak of SARS-CoV-2 in a multi-family residential building in Santander, Spain. The authors present a compelling case that a shared vertical bathroom ventilation duct (shunt system) served as the conduit for airborne transmission, infecting 15 residents across four apartments. The study employs a robust multi-disciplinary methodology, combining epidemiological tracing, whole-genome sequencing, field environmental measurements, CFD simulations, and CONTAM multi-zone modeling. The findings highlight the critical role of building ventilation defects in disease transmission, particularly in older housing stock.

2. General Assessment

The manuscript is well-written, timely, and addresses a significant intersection between public health and building engineering. The integration of genetic sequencing with engineering airflow analysis provides a high level of credibility to the conclusions.

Strengths:

• Genetic Confirmation: The use of genomic epidemiology effectively rules out community transmission. The authors note:

"The sequences derived from the building were similar, with one or two nucleotide differences among them and at least 11 differences from the closest control sequence." This strongly supports the hypothesis of a single internal source.

• "Negative" Evidence: The observation that apartments with modified ventilation systems were spared provides excellent real-world validation of the proposed transmission mechanism.

"Occupants of three homes... in which the bathroom ventilation had been modified (by installing an exhaust fan with a no-return flap) did not get infected."

However, there are specific areas regarding the generalization of results and the discussion of "control" areas within the building that require clarification before publication.

3. Specific Comments & Critique

A. The "Control Group" and Building-Wide Generalization

Critique: The study focuses heavily on the infected vertical stack (Home A/B stack). While this is logical for forensic analysis, the manuscript lacks physical measurements from the other three courtyards (patios) in the building that did not experience outbreaks.

The authors state:

"No positive PCR tests were detected among the residents of the homes surrounding the three other patios of the building."

However, regarding physical measurements, it is noted:

"Additional measurements in other homes were not possible due to a lack of access."

Recommendation: The authors must address this limitation more explicitly in the Discussion section. A critical reader might ask: "Did the other courtyards not have outbreaks because their ventilation worked better, or simply because there was no virus present?" Based on the architectural symmetry implied in the text ("The building... comprises seven floors with eight homes per level, grouped around four patios"), it is scientifically sound to assume the airflow physics are identical. The authors should explicitly state that the lack of infection in other courtyards is likely due to the absence of an Index Case in those stacks, rather than superior ventilation performance. This distinction is crucial for generalizing the risk to the entire building type, not just the specific duct investigated.

B. The Exacerbating Role of Kitchen Hoods

Critique: One of the most significant findings of this study—which has profound public health implications—is the counter-intuitive impact of kitchen exhaust fans. The data suggests that operating these fans can be more detrimental than passive ventilation issues.

The manuscript states:

"airflow into the upper bathroom reaches approximately 2 m/s, and CO2 levels are higher. These results indicate that operating a kitchen hood may enhance the risk of aerosol transmission... more than opening the patio window."

Furthermore, the mechanism is clearly identified:

"When exhaust rates are high and makeup air is insufficient, the resulting depressurization may draw air and potentially contaminants from adjacent areas."

Recommendation: This finding deserves greater prominence in the Abstract and Conclusions. The authors should formulate a specific recommendation for residents in buildings with shared ducts: utilizing powerful kitchen extraction without ensuring adequate make-up air (e.g., opening a street-facing window) can actively suck pathogens from neighbors' apartments.

C. Generalization from Single-Floor Measurements

Critique: The environmental boundary conditions for the models were derived from a single apartment on the 4th floor.

"Environmental measurements were collected in the 4th-floor bathroom... of Home B."

Recommendation: While the CFD and CONTAM models extrapolate this to other floors, the authors should briefly acknowledge in the Limitations section that "stack effect" pressure differentials vary by height. Therefore, the infiltration rates on the 1st or 2nd floor might differ slightly in magnitude from the measured 4th floor, though the direction of flow (reverse flow) likely remains consistent under the observed conditions.

D. CFD Model Simplification

Critique: The authors mention simplifying the geometry for the simulation.

"The model focused on airflow and CO2 distribution... with some non-essential features simplified to reduce computation time."

Recommendation: Please verify or briefly confirm in the text that these "non-essential features" do not include roughness elements or duct junctions that might significantly induce turbulence or pressure drops, which could affect the vertical propagation speed of the aerosols.

4. Conclusion

This paper provides valuable evidence supporting airborne transmission via shared infrastructure. The methodology is sound, and the conclusions are supported by the data. I recommend Minor Revisions to address the discussion points regarding the unmeasured courtyards and to strengthen the warnings regarding kitchen exhaust operation.

Status: Accept with Minor Revisions.

6. PLOS authors have the option to publish the peer review history of their article (what does this mean?). If published, this will include your full peer review and any attached files.

Reviewer #1: No

Reviewer #2: No

---

## [Author Response · Author response to Decision Letter 1]

2 Feb 2026

To: External Peer Review Report – PLOS One Manuscript

From: Shelly Miller and co-authors

Date: 1/28/2026

Re: Response Memo to comments on PONE-D-25-62493 Title: “Potential Airborne Transmission of SARS-CoV-2 Through Bathroom Ventilation Ducts Associated with an Outbreak in a Residential Building in Santander, Spain, 2020”

Recommendation: Minor Revisions

Thank you very much for the detailed comments on our submitted manuscript. We are grateful that you appreciated our interdisciplinary work and recommended minor revisions. Your suggestions have helped us to significantly improve and clarify our work.

Below we provide the Comments from Reviewer in grey font that require our response and edits to the manuscript. Our response is in italic black font and the changes to the manuscript are copied here in blue font.

Comments from Reviewer:

3. Specific Comments & Critique

A. The "Control Group" and Building-Wide Generalization

Critique: The study focuses heavily on the infected vertical stack (Home A/B stack). While this is logical for forensic analysis, the manuscript lacks physical measurements from the other three courtyards (patios) in the building that did not experience outbreaks.

The author’s state:

"No positive PCR tests were detected among the residents of the homes surrounding the three other patios of the building."

However, regarding physical measurements, it is noted:

"Additional measurements in other homes were not possible due to a lack of access."

Recommendation: The authors must address this limitation more explicitly in the Discussion section. A critical reader might ask: "Did the other courtyards not have outbreaks because their ventilation worked better, or simply because there was no virus present?" Based on the architectural symmetry implied in the text ("The building... comprises seven floors with eight homes per level, grouped around four patios"), it is scientifically sound to assume the airflow physics are identical. The authors should explicitly state that the lack of infection in other courtyards is likely due to the absence of an Index Case in those stacks, rather than superior ventilation performance. This distinction is crucial for generalizing the risk to the entire building type, not just the specific duct investigated.

Response: Thank you for this insightful comment. Your comment is accurate: no index case existed in the other courtyards. Furthermore, all the four courtyards are identical in architecture and have the same bathroom exhaust configuration as the case studied in this article. We agree that we must explicitly state that the lack of infection was likely because there was no index case in those areas of the building. Thus, we have added the following statement on Page 8, under the Testing section:

“The lack of infection around the other courtyards is likely due to the absence of an Index Case in those homes. PCR tests were conducted in the whole building (not only to the A and B sections of the building; Fig. 2).”

B. The Exacerbating Role of Kitchen Hoods

Critique: One of the most significant findings of this study—which has profound public health implications—is the counter-intuitive impact of kitchen exhaust fans. The data suggests that operating these fans can be more detrimental than passive ventilation issues.

The manuscript states:

"airflow into the upper bathroom reaches approximately 2 m/s, and CO2 levels are higher. These results indicate that operating a kitchen hood may enhance the risk of aerosol transmission... more than opening the patio window."

Furthermore, the mechanism is clearly identified:

"When exhaust rates are high and makeup air is insufficient, the resulting depressurization may draw air and potentially contaminants from adjacent areas."

Recommendation: This finding deserves greater prominence in the Abstract and Conclusions. The authors should formulate a specific recommendation for residents in buildings with shared ducts: utilizing powerful kitchen extraction without ensuring adequate make-up air (e.g., opening a street-facing window) can actively suck pathogens from neighbors' apartments.

Response: Thank you for noticing the importance of this issue. We agree that the use of the kitchen exhaust system deserves a little more attention and focus on the manuscript. We also believe that the work presented in this article will hopefully expand the knowledge and design of proper building ventilation design to consider disease transmission. To address this comment, we have added the following to the Abstract:

“Additionally, operating the kitchen exhaust fan can augment the movement of aerosols between occupied spaces increasing the potential for infection.”

We have added the following to the Conclusions:

“Additionally, in buildings with shared ducts, sufficient make up air is needed to prevent kitchen exhaust use from drawing air—potentially containing pathogens—from neighboring apartments. This can be achieved by opening a street-facing window.”

C. Generalization from Single-Floor Measurements

Critique: The environmental boundary conditions for the models were derived from a single apartment on the 4th floor.

"Environmental measurements were collected in the 4th-floor bathroom... of Home B."

Recommendation: While the CFD and CONTAM models extrapolate this to other floors, the authors should briefly acknowledge in the Limitations section that "stack effect" pressure differentials vary by height. Therefore, the infiltration rates on the 1st or 2nd floor might differ slightly in magnitude from the measured 4th floor, though the direction of flow (reverse flow) likely remains consistent under the observed conditions.

Response: Thank you for this comment. Very good point. We have added the following text to the CFD model section on page 14:

“Note that because "stack effect" pressure differentials vary by height, the infiltration rates on the 1st or 2nd floor might differ in magnitude from the measured 4th floor, though the direction of flow (reverse flow) likely remains consistent under the observed conditions.”

We did not add any text to the CONTAM model section because we did not use the measurements from the 4th floor apartment to derive the model. We used the measurements to verify that the model output was accuracy. A key strength of our study is that the CONTAM model for the Santander building captures real-world multizone airflow, including the stack effect within the shared vertical shaft, enabling us to evaluate aerosol transmission throughout the building.

D. CFD Model Simplification

Critique: The authors mention simplifying the geometry for the simulation.

"The model focused on airflow and CO2 distribution... with some non-essential features simplified to reduce computation time."

Recommendation: Please verify or briefly confirm in the text that these "non-essential features" do not include roughness elements or duct junctions that might significantly induce turbulence or pressure drops, which could affect the vertical propagation speed of the aerosols.

Response: Thank you so much for this insightful and critical comment. This is correct, as the conditions for our CFD simplified model include the calculated duct roughness (based on the typical roughness of brick and mortar, the duct material used in this building). The duct is a vertical shaft made of brick and mortar, and no duct junctions are implemented in this type of construction. We have added the following statement to Page 14, under the CFD section:

“This simplified CFD model includes variables such as duct roughness based on the typical roughness average of brick and mortar which is the material used for the construction of the bathroom exhaust.”

4. Conclusion

This paper provides valuable evidence supporting airborne transmission via shared infrastructure. The methodology is sound, and the conclusions are supported by the data. I recommend Minor Revisions to address the discussion points regarding the unmeasured courtyards and to strengthen the warnings regarding kitchen exhaust operation.

Status: Accept with Minor Revisions.

Please let us know if you have any additional questions or comments. Thank you again for your time and your insights provided in this peer review.

Respectfully,

Shelly Miller

---

## [Decision Letter · Decision Letter 1]

2 Mar 2026

Potential Airborne Transmission of SARS-CoV-2 Through Bathroom Ventilation Ducts Associated with an Outbreak in a Residential Building in Santander, Spain, 2020

PONE-D-25-62493R1

Dear Dr. Miller,

We’re pleased to inform you that your manuscript has been judged scientifically suitable for publication and will be formally accepted for publication once it meets all outstanding technical requirements.

Kind regards,

Baohua Wen

Academic Editor

PLOS One

Additional Editor Comments (optional):

Reviewers' comments:

Reviewer's Responses to Questions

**Comments to the Author**

1. If the authors have adequately addressed your comments raised in a previous round of review and you feel that this manuscript is now acceptable for publication, you may indicate that here to bypass the “Comments to the Author” section, enter your conflict of interest statement in the “Confidential to Editor” section, and submit your "Accept" recommendation.

Reviewer #1: All comments have been addressed

Reviewer #2: All comments have been addressed

2. Is the manuscript technically sound, and do the data support the conclusions?

Reviewer #1: (No Response)

Reviewer #2: Yes

3. Has the statistical analysis been performed appropriately and rigorously? 

Reviewer #1: (No Response)

Reviewer #2: Yes

4. Have the authors made all data underlying the findings in their manuscript fully available?

Reviewer #1: (No Response)

Reviewer #2: Yes

5. Is the manuscript presented in an intelligible fashion and written in standard English?

Reviewer #1: (No Response)

Reviewer #2: Yes

6. Review Comments to the Author

Reviewer #1: (No Response)

Reviewer #2: (No Response)

7. PLOS authors have the option to publish the peer review history of their article (what does this mean?). If published, this will include your full peer review and any attached files.

Reviewer #1: **Yes:** Pari Alavi

Reviewer #2: **Yes:** Abdullah Al_Sarraf

---

## [Editor Report · Acceptance letter]

PONE-D-25-62493R1

PLOS One

Dear Dr. Miller,

I'm pleased to inform you that your manuscript has been deemed suitable for publication in PLOS One. Congratulations! Your manuscript is now being handed over to our production team.

Kind regards,

on behalf of

Dr. Baohua Wen

Academic Editor

PLOS One